# WHY CONTEXT MATTERS IN VQA & REASONING:
# SEMANTIC INTERVENTIONS FOR VLM INPUT MODALITIES

## ABSTRACT

The various limitations of Generative AI, such as hallucinations and model failures, have made it crucial to understand the role of different modalities in Visual Language Model (VLM) predictions. Our work investigates how the integration of information from image and text modalities influences the performance and behavior of VLMs in visual question answering (VQA) and reasoning tasks. We measure this effect through answer accuracy, reasoning quality, model uncertainty, and attention attribution to each modality. We study the interplay between text and image modalities in different configurations where visual content is essential for solving the VQA task. Our contributions include (1) the Semantic Interventions (SI)-VQA dataset, (2) a benchmark study of various VLM architectures under different modality configurations, and (3) the Interactive Semantic Interventions (ISI) tool. The SI-VQA dataset serves as the foundation for the benchmark, while the ISI tool provides an interface to test and apply semantic interventions in image and text inputs, enabling more fine-grained analysis. Our results show that complementary information between modalities improves answer and reasoning quality, while contradictory information harms model performance and confidence. Image text annotations have minimal impact on accuracy and uncertainty, slightly increasing image relevance. Attention analysis confirms the dominant role of image inputs over text in VQA tasks. In this study, we evaluate state-of-the-art VLMs that allow us to extract attention coefficients for each modality. A key finding is PaliGemma's harmful overconfidence, which poses a higher risk of silent failures compared to the LLaVA models. This work sets the foundation for rigorous analysis of modality integration, supported by datasets specifically designed for this purpose. The code is available at `https://gitlab.com/dekfsx1/si-vlm-benchmark` and the tool and dataset are hosted at `https://gitlab.com/dekfsx1/isi-vlm`.

## 1 INTRODUCTION

Vision-Language Models (VLMs) have shown remarkable performance across various NLP tasks by combining visual and textual information. Models like LLaVA (Liu et al., 2023a), GPT4-Vision (OpenAI, 2024), and PaliGemma (Beyer et al., 2024) use the text interface of Large Language Models (LLMs) alongside CLIP-style image encoders (Radford et al., 2021), making them well-suited for multi-modal tasks such as content summarization (Moured et al., 2024), text-guided object detection (Dorkenwald et al., 2024), and visual question answering (VQA) (Yue et al., 2024b).

The core principle of VLMs lies in the integration and interplay of the two modalities, which significantly amplifies the effectiveness of the models. However, a recurrent problem for multimodal training and testing is the modality bias, i.e., when a modality dominates another in a multimodal setting (Guo et al., 2023; Liang et al., 2022). The extent to which each modality influences the final answer and reasoning of VLMs remains unclear within the research community, with varying and sometimes conflicting conclusions (Gat et al., 2021; Frank et al., 2021; Chen et al., 2024). Understanding how modalities impact VLMs by exploring complex and specific scenarios is crucial as multimodal models become increasingly opaque and diverse. This approach would not only aid in understanding and mitigating model failures, such as hallucinations but also guide researchers toward practical solutions for improving model performance and self-consistency.

To this end, we present a benchmark, investigating the effect of semantic interventions on text and image modalities, alongside a carefully curated dataset and an interactive tool. Our work demonstrates how task-specific modality configurations can reveal heterogeneous effects on model performance and uncertainty. Specifically, we investigate the role of the text modality as a context guide in VQA tasks. We make the image the necessary data source to answer the question while using the text in different configurations to affect the model's answer and reasoning. Initially, we develop the **Interactive Semantic Interventions (ISI) Tool** to perform inter-modality interventions, testing their effect

---

*Contributed Equally

on the model behavior and observing which interventions lead to model failure. Based on the results, we curate the **Semantic Interventions (SI-)VQA dataset**. It contains 100 cautiously crafted examples with controlled interventions in both the input image and context. Each instance includes an image, the same image annotated, a complementary or contradictory textual context, and a question with a ground truth Yes/No answer, which can only be inferred from the image. This dataset allows for a wide range of image-text combinations. Using the SI-VQA Dataset, we assess the impact of various interventions on VLMs across multiple evaluation criteria. We experiment with different configurations where text serves as either complementary or contradictory context to the image, or as annotations directly on the image. We then analyze the model's performance, uncertainty, and attention attribution across these modalities. Specifically, we examine the latest open-source VLMs including LLaVA 1.5 (Liu et al., 2023a), LLaVA-Vicuna (Liu et al., 2024a), LLaVA-NeXT (Liu et al., 2024b), and PaliGemma (Beyer et al., 2024). Our work focuses on probing the behavior of large vision-language models (VLMs) in controlled multimodal scenarios, rather than benchmarking their performance or proposing a new large-scale multimodal benchmark for state-of-the-art results. Additionally, we do not aim to address modality bias but instead explore the roles and interactions of modalities to better understand VLMs' strengths and limitations.

Our study demonstrates that integrating complementary contextual information into VQA models enhances both their answer accuracy and reasoning quality, while contradictory information significantly degrades performance and confidence, comparable to the absence of visual input. Additionally, we find that these models inherently prioritize visual data over textual context; efforts to rebalance attention toward textual information—such as adding image annotations, textual descriptions, or prompt engineering—yield mixed results, with prompt engineering improving accuracy without altering attention distribution and textual descriptions unexpectedly reducing accuracy and increasing uncertainty. Using the AUGRC score to measure the frequency of silent failures, our experiments reveal the harmful overconfidence of the PaliGemma 3B model compared to the LLaVA models. These findings provide valuable insights for the AI community by highlighting the critical role of context in VQA models, guiding future developments to optimize attention between visual and textual modalities, and assessing the risk of silent model failures.

In summary, our contributions are:

- A benchmark to evaluate the semantic interplay between image and textual context based on diverse evaluation criteria and several state-of-the-art VLMs, meant to facilitate silent failure detection.

- Ablation studies regarding rebalancing attention to specific modalities and the effect of model quantization.

- The SI-VQA Dataset, a well-curated dataset that enables the exploration of various combinations of image and text modalities. The benchmark and dataset are hosted at: `https://gitlab.com/dekfsx1/si-vlm-benchmark`.

- The ISI Tool, a tool designed to enable users to investigate how VLMs respond to semantic changes and interventions across image and text modalities, with a focus on identifying potential model failures. The tool is hosted at: `https://gitlab.com/dekfsx1/isi-vlm`.

## 2 RELATED WORK

**VQA Datasets**   Since the introduction of the Visual Question Answering (VQA) task in Antol et al. (2015), numerous datasets have been created to support research in this area, such as GQA (Hudson & Manning, 2019), Visual7W (Zhu et al., 2016), Visual Genome (Krishna et al., 2017), CLEVR (Johnson et al., 2017), and VQA 2.0 (Goyal et al., 2017). However, these datasets often lack reasoning, are restricted to textual-only modalities, or suffer from small-scale and limited domain diversity. ScienceQA (Lu et al., 2022) emerged as the first large-scale VQA dataset to incorporate textual context alongside textual explanations for model reasoning. It features multiple-choice questions across various scientific domains, with each question annotated with lectures and explanations. More diverse multimodal datasets, such as SEED (Li et al., 2023), MMBench (Liu et al., 2023b), and MMMU (Yue et al., 2024a), have since been introduced, combining text and image data. However, as shown by Chen et al. (2024), these datasets do not ensure that all evaluation samples require visual content for correct answers. To address this limitation, we propose a new curated VQA dataset designed such that answers can only be derived from the image, with complementary or contradictory context provided as additional information to influence the model's answer and reasoning.

**Modalities in Vision-Language Tasks**   Several methods have been proposed to investigate the extent to which VLMs leverage both visual and textual information. *Annotation and foiling approaches* introduce text annotation in images and mistakes in image descriptions and test whether the VLM predictions change. Shekhar et al. (2019); Parcalabescu et al. (2022) test VLMs' sensitivity to discrepancies between images and captions and found that models often overlook such inconsistencies. (Gat et al., 2021) exchange images and captions with other instances and note a consistent

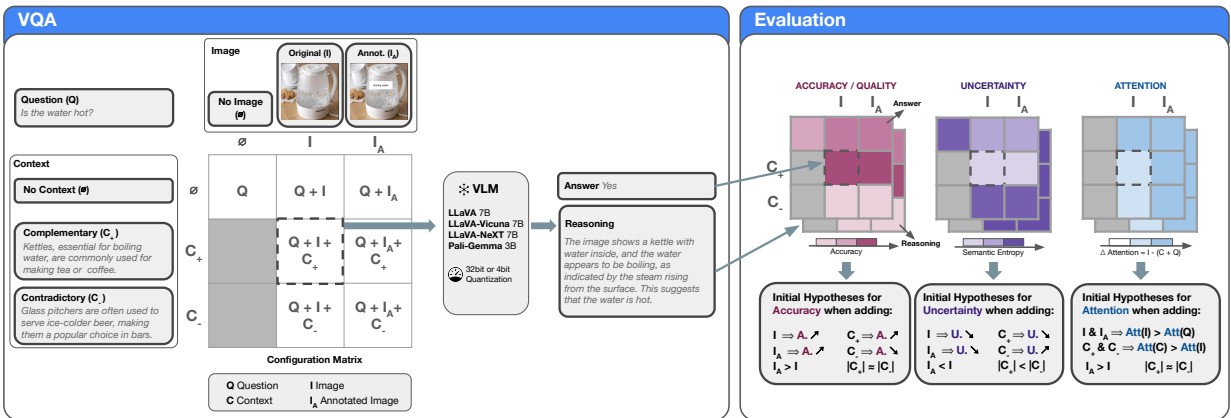

Figure 1: The SI-VQA framework examines the influence of various modality configurations on **answer accuracy and reasoning quality**, model **uncertainty**, and **attention attribution**. Seven different configurations are tested, combining inputs such as the question (Q), image (I), annotated image ($I_A$), and either complementary ($C_+$) or contradictory context ($C_-$): (Q), (Q+I), (Q+I+$C_+$), (Q+I+$C_-$), (Q+$I_A$), (Q+$I_A$+$C_+$), and (Q+$I_A$+$C_-$). For each configuration, the VLM is assessed first on its answer and then on its reasoning. Furthermore, we establish prior assumptions regarding how each modality is expected to impact the model's behavior.

decrease in accuracy, with textual input proving to be more influential than visual content. Building on these findings, we incorporate textual annotations into images in our study to explore how textual data, shown in previous work to be more significant than pixels, can support model reasoning. *Ablation methods* investigate model behavior when parts of the input are removed or masked (Bugliarello et al., 2021; Cafagna et al., 2021). Frank et al. (2021) occludes parts of the image or masks the text, and finds that the visual modality matters more than the text. In line with this, we introduce semantic perturbations, creating scenarios where the text complements the image in various ways. *Attention-based methods* correlate high attention scores with high feature importance, though this connection remains debated. (Serrano & Smith, 2019; Jain & Wallace, 2019) questioned the validity of attention as an indicator of importance, while (Wiegreffe & Pinter, 2019) challenged their arguments, asserting that attention can serve as an explanation, albeit not the definitive one. Following mechanistic interpretability, our paper also analyzes how attention scores are attributed across modalities in different text-image configurations. We view attention attribution as one key factor in understanding the intrinsic roles of each modality in VLM performance. Early work on the impact of modalities in VQA has produced mixed results. While some studies argue that VLMs rely more on text (Gat et al., 2021), others emphasize the importance of the visual modality (Frank et al., 2021). More recently, Chen et al. (2024) showed that visual data might be unnecessary to answer several questions, suggesting among others unintentional data leakage through LLM and VLM training. Prior research has highlighted the dominance of textual data (Parcalabescu & Frank, 2024b) for Vision Language decoders. Our paper builds on this debate by proposing a rigorous framework for conducting semantic interventions on both modalities to assess their respective contributions to VQA tasks.

**Multimodal XAI** Multimodal Explainable AI (MXAI) is a branch of XAI that includes a range of XAI techniques tailored to address the unique challenges posed by multimodal data inputs, tasks, and architectures. Recent reviews study the new challenges and differences of those methods as compared to traditional XAI approaches (Joshi et al., 2021; Rodis et al., 2023). In the multimodal context, MXAI leverages the complementary explanatory strengths of different modalities offering richer insights. In specific scenarios, language can provide a deeper understanding and clarification of concepts, while in others, the visual modality may be more informative (Park et al., 2018). MXAI is used as a tool to interpret VQA tasks that might require higher-order reasoning and a deep understanding of semantic context (Antol et al., 2015).

## 3 SI-VQA DATASET AND ISI TOOL

This section introduces the dataset and tool developed to examine the role of modalities in VQA & Reasoning. They are designed to explore semantic interventions on input modalities for thorough post-hoc interpretability of VLMs.

## 3.1 SI-VQA DATASET

The SI-VQA dataset is a closed-question VQA dataset consisting of 100 samples. Each sample consists of an image, question, and ground truth answer (Yes/No) pair, as well as one text annotated image, one contradictory context, and one complementary context. Unlike Chen et al. (2024), which uses multichoice questions, we choose a binary question format (Yes/No) to eliminate issues related to distractor quality and random guessing, ensuring more precise evaluations. The questions in SI-VQA dataset require understanding object significance and simple interactions in the image without reaching the higher complexity levels of datasets like visual commonsense reasoning (Zellers et al., 2019) or scene graph reasoning (Hildebrandt et al., 2020). The image is the necessary and sufficient element to get the question right as it contains the critical information. Thus, the model has a very limited ability to leverage prior knowledge to answer the question and must utilize the image input, avoiding the data leakage in SOTA VQA datasets such as ScienceQA, AI2D, MMMU, and SEED (Chen et al., 2024). The context is always in relation to the ground truth answer, aiming to either confuse or help the model without stating an explicit answer to the question, as each question can only be answered through the image. The image annotations provide text-based hints to the ground truth answer while avoiding annotation artifacts (Zellers et al., 2019), ensuring that the model's attention to the question remains constant regardless of the presence of annotations. We only study the impact of "positive" annotations, i.e., text that informs the model about key elements in the image to correctly answer the question. See Figure 1 for an exemplary sample. The images are open-source and from the MMMU Benchmark (Yue et al., 2024b). The selected images of SI-VQA Dataset span a variety of fields, including geography, history, art and design, sport, and biology, along with everyday objects and landscapes. This ensures that SI-VQA maintains a comparable diversity of topics to existing SOTA datasets. They encompass diverse formats, such as natural photographs, cartoons, sketches, and paintings. We carefully crafted all other modalities for the dataset, focusing on quality not quantity.

We design seven scenarios for VLM interpretability analysis, creating seven modality configurations: question (Q), question + image (Q+I), question + image + complementary context (Q+I+C$_+$), question + image + contradictory context (Q+I+C$_-$), question + annotated image (Q+I$_A$), question + annotated image + complementary context (Q+I$_A$+C$_+$), question + annotated image + contradictory context (Q+I$_A$+C$_-$). They can be inferred using the 3x3 matrix in Figure 1. For the baseline configuration Q (question-only), the image corresponds to a black image. In this case, the model's answer accuracy is at random (See Appendix subsection H.1) proving the necessity of visual content in SI-VQA Dataset (Chen et al., 2024). We also conducted the baseline experiment using noise-only images and with the images entirely removed, observing similar results in both cases (See Appendix C).

## 3.2 ISI TOOL

To provide an intuitive and agile way to explore modality interplay in the context of multimodal interpretability, we developed the ISI Tool. We used this interactive tool to design the SI-VQA Dataset. It is designed to enable researchers and VLM users to investigate how VLMs respond to semantic changes and interventions across image and text modalities, with a focus on identifying potential model failures in the context of VQA. Specifically, it allows the perturbation of images, the addition of personalized shapes and annotations, and the arbitrary adaptation of text inputs. Users can upload their own images and questions or choose from 100 preloaded samples with semantic intervention presets from our ISI-VQA Dataset. More details about the tool and its features are available in Appendix E.

## 4 EXPERIMENT METHODOLOGY

Our benchmark pipeline illustrated in Figure 1 investigates the contribution of each modality toward the performance of several VLMs. Performance is measured in terms of output quality, model uncertainty, and attention attribution toward the input elements of the SI-VQA Dataset.

### 4.1 VISION-LANGUAGE MODELS

The VLMs selected for this study are state-of-the-art models for Visual Question Answering (VQA) tasks. We excluded models where extracting attention coefficients for each modality is not feasible, such as Flamingo (Alayrac et al., 2022), which employs gated cross-attention between text and image. The final architectures chosen for evaluation are LLaVA 1.5, LLaVA-Vicuna, LLaVA-NeXT, and PaliGemma (Beyer et al., 2024), with weights provided from HuggingFace (Wolf et al., 2020). LLaVA-Vicuna is a version of LLaVA 1.5 leveraging the Vicuna LLM, a conversation-fine-tuned version of LLaMA. LLaVA-Vicuna and LLaVA-NeXT both utilize dynamic high resolution for the image input (Liu et al., 2024a), increasing visual reasoning and optical character recognition (OCR) capabilities. Although PaliGemma is intentionally designed for pre-training followed by fine-tuning, we employ it in this study

within a zero-shot setting. We do not present any reasoning results for the PaliGemma model as it mainly generates a default response "Sorry, as a base VLM I am not trained to answer this question." when it is asked to explain its answer due to its strong safety fine-tuning (Beyer et al., 2024). Each LLaVA architecture comprises 7B parameters, while PaliGemma consists of 3B parameters, and all can be quantized to reduce VRAM usage and improve computational efficiency. We restrict our analysis to 7B models because this number of parameters aligns with the difficulty of our VQA dataset and provides an optimal balance for meaningful performance variation. In addition, our ablation study in Appendix F shows that, while 32bit models have lower uncertainty in VQ answering and reasoning, answer accuracy is not substantially worse for even 4bit quantized models.

## 4.2 ANSWER & REASONING EVALUATION

We assess the model's performance on the VQA task using the SI-VQA Dataset by evaluating both its answer and reasoning. To measure the VQ answering quality, we use the accuracy metric by comparing the model's binary Yes/No response to the closed question with the ground truth provided in the dataset. Evaluating reasoning is more complex, as no ground truth exists for the explanations. Without a reference rationale, we assess reasoning based on the quality of argumentation and the truthfulness of statements. To this end, we use GPT-4o as an external evaluator (Zheng et al., 2024; Thakur et al., 2024). The model is prompted once for each sample to rate the reasoning from 0 to 10, considering an evaluation prompt as well as the question, image, answer, and reasoning. While the quality scores seem very reasonable to us, we observe a bias toward the score number "8" (see subsection H.1), a behavior also observed in other studies using LLMs as a judge (Thakur et al., 2024). See Appendix B for the evaluation prompts and hyperparameters.

## 4.3 MODEL UNCERTAINTY

$$SE(x) = -\sum_c p(c|x) \log p(c|x) = -\sum_c \left( \left( \sum_{s \in c} p(s|x) \right) \log \left[ \sum_{s \in c} p(s|x) \right] \right) \tag{1}$$

For quantifying model uncertainty, we employ semantic entropy $SE()$ (Farquhar et al., 2024), which calculates entropy based on the sum of token likelihoods $p()$ between the sets $c$ of semantically similar clustered sentences $s$ (see Equation 1). For semantic clustering, we use the DeBERTa (He et al., 2021) entailment model. During uncertainty computation, the number of sampled outputs and the sampling temperature $T$ are set to 10 and 0.9 respectively. High $SE()$ means high uncertainty and low confidence in the outputs.

For VLMs, uncertainty quantification is especially important for detecting model failures, including hallucinations and silent failures. Silent failures are instances where the model generates incorrect information with high confidence, making these errors increasingly difficult to detect (Bender et al., 2021; Jaeger et al., 2023). The Area Under the Generalized Risk Coverage curve (AUGRC) metric (Traub et al., 2024) evaluates the extent to which a model makes incorrect predictions with high confidence, where high confidence is characterized by low semantic entropy, specifically below a defined threshold, $\tau$:

$$\text{AUGRC} = \int_0^1 P(Y_\text{f} = 1, \, SE(x) \le \tau) \, \mathrm{d}P(SE(x) \le \tau) \tag{2}$$

With $Y_\text{f}$ as the binary failure indicator, i.e., $Y_\text{f} = 1$ indicates a wrong prediction by the model. We refer to Traub et al. (2024) for a detailed explanation. The AUGRC therefore directly measures the harmful overconfidence of a model, i.e., when the model is confident and wrong. A high AUGRC indicates that the model has a tendency for silent failures.

## 4.4 ATTENTION ATTRIBUTION

Attributing attention values to the different modalities serves as one indicator of each modality's contribution to the final answer and reasoning (Wiegreffe & Pinter, 2019). We adopt a mechanistic approach, treating attention as a VLM interpretability measure to evaluate the roles of the question, image, and context. By aggregating attention across layers and heads, we derive relevance scores for each token's contribution to the answer or reasoning (Parcalabescu & Frank, 2023; 2024a). To determine which input modality is most relevant to the prediction, we sum the relevance scores of their respective input tokens. Finally, to standardize the scores for comparability, we compute the relative relevance score for each sample. A detailed explanation of attention aggregation and implementation is provided in Appendix A.

# 5 EXPERIMENT RESULTS

## 5.1 INITIAL HYPOTHESES

Based on the seven modality configurations and the results of previous related work, we define three prior hypotheses regarding the anticipated outcomes when including a new type of input. As discussed in subsection 4.1, only the VQ answering results are shown for PaliGemma.

***Hypothesis 1 (Including Image):*** In the SI-VQA dataset, images are essential for VQA tasks, as they serve as the only modality providing the necessary information to answer the questions. Therefore, we hypothesize a significant positive impact on accuracy and a reduction in model uncertainty when the image is incorporated alongside the question. Furthermore, the natural image is expected to receive greater attention compared to the baseline black or random pixel images in the question-only configuration. The impact on reasoning quality, however, remains in our opinion uncertain.

***Hypothesis 2 (Including Context):*** We hypothesize that adding complementary context will improve model accuracy, confidence, and reasoning quality. This additional information is expected to facilitate more detailed and precise rationalization. Conversely, contradictory context is anticipated to decrease model accuracy, confidence, and reasoning quality, reflecting the model's doubt. Based on previous research, we also expect the model to exhibit a higher attention toward text compared to the image.

***Hypothesis 3 (Including Image Annotations):*** Compared to configurations including the natural image, those utilizing the annotated image are expected to enhance the model's ability to answer and reason with greater accuracy and confidence. Additionally, we anticipate increased attention toward the annotated image relative to the natural image.

In the following sections, we will first present the benchmark results before comparing them to the prior hypotheses we have just put forward.

## 5.2 ANSWER & REASONING EVALUATION

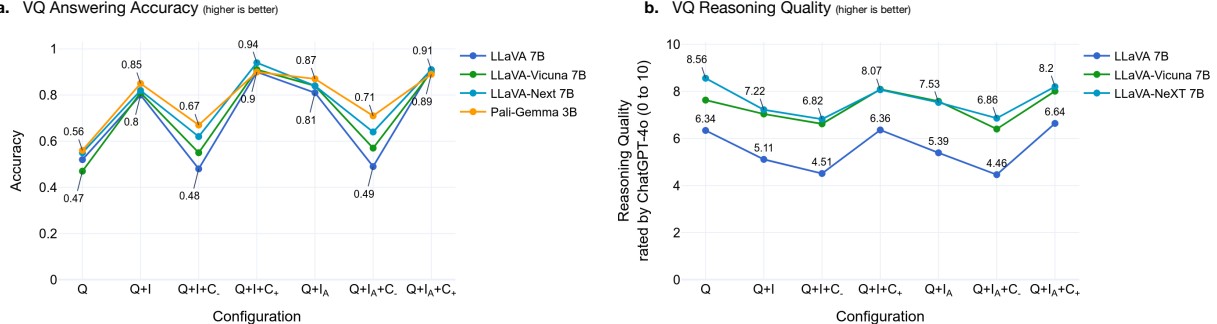

Figure 2: Quality of VLM answers and reasoning in the seven modality configurations of question (Q), image (I), annotated image ($I_A$), complementary ($C_+$) and contradictory context ($C_-$). Answer accuracy is measured using the ground truth labels of our SI-VQA Dataset and reasoning quality is evaluated using the external scoring of GPT-4o as a judge. A significant drop of accuracy in the answer and reasoning is observed for all models when adding contradictory context, i.e., Q+I+$C_-$ and Q+$I_A$+$C_-$. Results for PaliGemma 3B are only displayed for answering (see subsection 4.1).

Figure 2 presents the results of the VQA accuracy (a.) and the quality of the VLM VQ reasoning (b.), as judged by GPT-4o. Similar patterns emerge across all models. The answer accuracy is low in the question-only baseline, where the model lacks sufficient information to provide correct answers. However, when given just a question and a black image, the models consistently offer strong reasoning quality, consistently justifying their response by acknowledging the absence of image information. Incorporating complementary context into the I+Q configuration enhances both answer accuracy and reasoning quality by providing additional details necessary for a correct response and a well-supported rationale. In contrast, the introduction of contradictory context significantly degrades response accuracy. The decline in accuracy is the smallest for PaliGemma, whereas for LLaVA, it drops to a level comparable to the question-only configuration. Additionally, reasoning quality declines as the models are misled by the conflicting information. When exchanging the natural image with an annotated image, we observe no change in accuracy or reasoning quality, even for architectures optimized for OCR.

When comparing the VLM architectures, a notable discrepancy emerges in their handling of contradictory information, with models responding differently to contradictions (configuration Q+I+$C_-$). Surprisingly, PaliGemma demonstrates

the most robustness in managing contradictions and achieving the highest accuracy scores in five out of seven configurations, despite having less than half the parameters of the LLaVA models and not being explicitly fine-tuned for VQA tasks. LLaVA-NeXT ranks second in accuracy but does not fully leverage its enhanced OCR capabilities when the annotated image is included. In terms of reasoning abilities, the conversational fine-tuned VLMs produce substantially higher-quality reasoning compared to the standard LLaVA 1.5 model.

> **Answer Accuracy & Reasoning Quality**
>
> While answer accuracy is low in the question-only baseline, models still provide strong reasoning quality by acknowledging missing image data. Complementary context improves accuracy and reasoning quality, but contradictory context significantly degrades performance. We observe the strongest decline for LLaVA and smallest in PaliGemma. Replacing the natural image with an annotated one shows no effect on accuracy or reasoning quality.

## 5.3 MODEL UNCERTAINTY

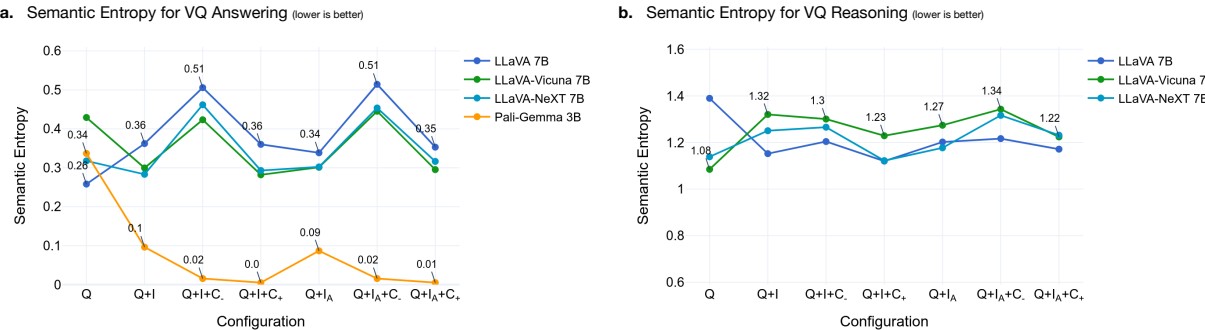

Figure 3: VLM uncertainty when generating answers and reasonings in the seven modality configurations of question (Q), image (I), annotated image ($I_A$), complementary ($C_+$) and contradictory context ($C_-$). Uncertainty is measured using the semantic entropy–the lower the entropy, the more confident the model. PaliGemma 3B shows extreme confidence overall in its answers. However, no reasoning results for PaliGemma are provided (see subsection 4.1). $C_-$ negatively impacts the certainty of LLaVA models when generating answers.

Figure 3 displays the model uncertainty measured through semantic entropy for both the answer (a.) and the reasoning (b.). For all models, the absence of image-based information (configuration Q) results in similar levels of uncertainty in both VQ answering and reasoning. In addition, we observe for all models that image text annotations have almost no impact on the model uncertainty compared to the configurations including the natural images.

For PaliGemma, adding image and context information significantly reduces uncertainty in VQ answering, making the model much more confident in its predictions. It seems that providing additional context, regardless of its content, leads the model to be more self-assured. This intriguing pattern shows large overconfidence in PaliGemma, which does not always have to be beneficial as it can, e.g., lead to silent failures, where the model is extremely confident in its wrong predictions (Bender et al., 2021; Jaeger et al., 2023).

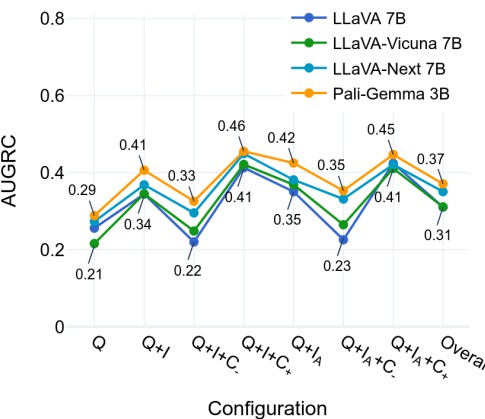

Figure 4: AUGRC evaluating the ability to detect silent failures through semantic entropy for each model and configuration (lower is better).

For all LLaVA models, we observe overall an inverse relationship between answer uncertainty and reasoning uncertainty, with LLaVA 1.5 exhibiting the highest uncertainty in VQ answering but the lowest in VQ reasoning.

When the image is added, LLaVA-Vicuna and LLaVA-NeXT show reduced uncertainty in VQ answering but increased uncertainty in VQ reasoning, as the models, in the question-only configuration, only acknowledge the absence of the image and therefore reason with high confidence. Complementary context slightly decreases model uncertainty, indicating a marginal increase in confidence for both VQ answering and reasoning. This effect is minor though, as

shown in Figure 3 b. where all LLaVA models exhibit nearly identical semantic entropies for I+Q and Q+I+C+, as well as for $Q+I_A$ and $Q+I_A+C+$. Contradictory contextual information, on the other hand, significantly increases uncertainty in the model answers. Its effect on reasoning is also particularly pronounced in LLaVA 1.5 but remains relatively minor for LLaVA-Vicuna and LLaVA-NeXT. Thus, the LLaVA models appear to be slightly influenced by reinforcing information sources but are more easily unsettled by contradictory ones.

**Model Failure Detection through Semantic Entropy** Interpreting VLMs' behavior through the lense of model uncertainty is crucial for identifying and understanding failures, including hallucinations and silent failures. Specifically, for PaliGemma, silent failures cannot be dismissed due to the model's extreme overconfidence, despite its prediction accuracy being comparable to that of LLaVA models. To quantify this harmful overconfidence—when the model confidently predicts incorrect answers with very low uncertainty—we employ the AUGRC metric, as detailed in Equation 2. Figure 4 displays the AUGRC values across all model architectures (where lower is better). Our results confirm that PaliGemma performs the worst, validating our hypothesis regarding its harmful overconfidence. Additionally, we observe that in cases of high uncertainty, such as with the $Q+I+C_-$ and $Q+I_A+C_-$ configurations, AUGRC is low, indicating fewer silent failures. In these scenarios, the contradictory context reduces the likelihood of confident incorrect answers, meaning the models become more uncertain about their mistakes, thereby making them more trustworthy.

> **Uncertainty**
>
> PaliGemma demonstrates high overconfidence, leading to silent failures, as evidenced by the AUGRC metric. LLaVA models display an inverse relationship between VQ answering and reasoning uncertainty, probably due to more detailed reasoning in more advanced models. Contradictory context significantly increases the VQ answering uncertainty for LLaVA models but has only a minor effect on VQ reasoning.

## 5.4 ATTENTION ATTRIBUTION

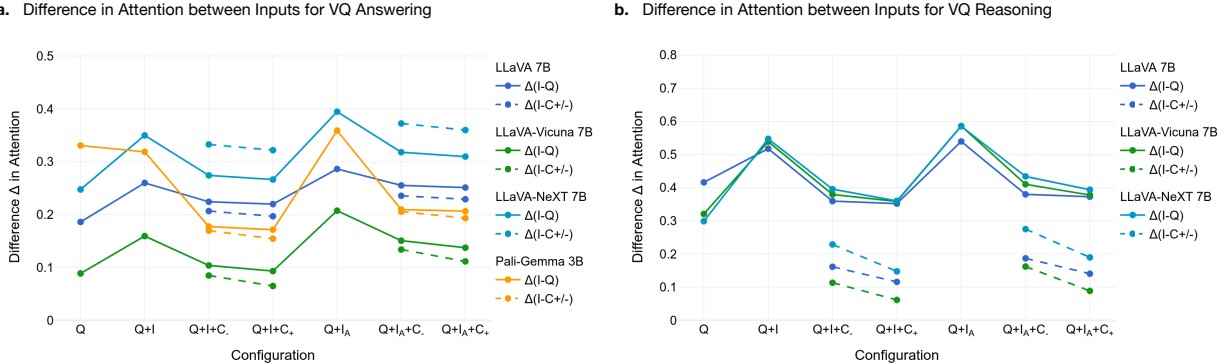

Figure 5: Difference in attention attribution between the image and the question (solid line) and between the image and the context when present (dashed line). The computation of modality attention attribution is described in Appendix A. The image (I) always gets the highest attention attribution compared to text modalities (Q, $C_-$, $C_+$). No reasoning results for PaliGemma are provided (see subsection 4.1).

This section examines how attention is distributed across the three inputs—question, image, and context—across the seven configurations. Figure 5 shows the average difference between attention to the image and the question, as well as the attention to the image and the context. Since all differences are positive, the image consistently receives the highest average attention in both VQ answering (Figure 5a.) and reasoning (Figure 5b.). The attention distribution across different inputs is similar among the models, with LLaVA-Vicuna showing the highest attention to textual inputs and LLaVA-NeXT focusing more on the image. Both answering and reasoning exhibit higher attention to the natural image compared to the black baseline image in the question-only configuration. Further, attention to the image decreases when context is added and the annotated image receives more attention than the natural image. In VQ answering, attention to the question and context is nearly equal, whereas, in VQ reasoning, the model shows significantly higher attention to the context, almost equal to the attention given to the image. Detailed figures are

provided in Appendix subsection H.3. Overall, no strong correlation is observed between attention attribution and accuracy (see Appendix subsection H.4).

> **Attention Attribution**
>
> The image modality receives the highest attention compared to the question and context, with its relevance further increasing when annotated. In VQ answering, attention to the question and context is nearly equal, whereas in VQ reasoning, the model allocates significantly more attention to the context. LLaVA-Vicuna pays the highest attention to textual inputs and LLaVA-NeXT to the image.

## 5.5 REBALANCING MODALITY IMPORTANCE

Given the observed high attention allocated to the image modality, it raises the question of how the results might change if we intervene to direct more attention toward the text modalities. Specifically, we aim to investigate how the model's performance changes when either the information in the image is described with text or the attention of the model is guided toward the text via prompt engineering.

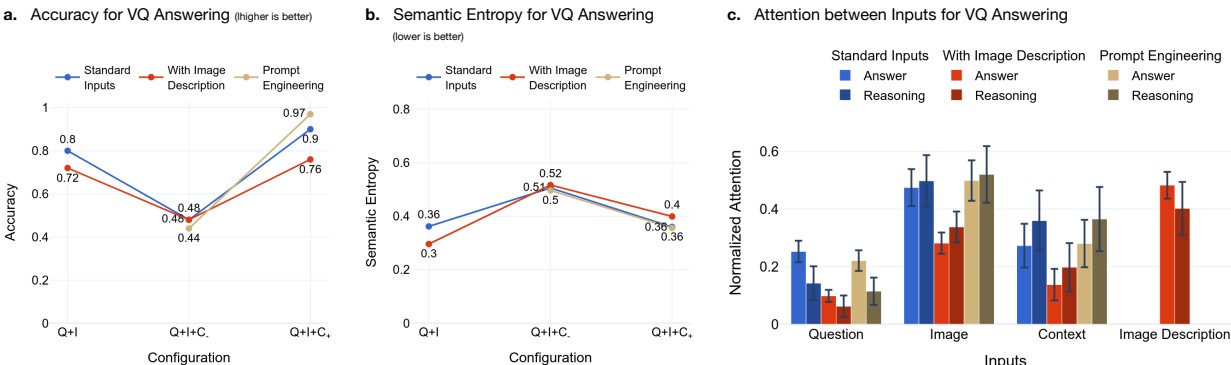

Figure 6: Two strategies to adjust modality importance: incorporating the image's textual description into the input and modifying the prompt to shift more attention toward the context. The impact of these changes is evaluated based on a. answer accuracy, b. model uncertainty, and c. attention attribution. The hereby experiments were conducted with the LLaVA 7B model, see Appendix I for the PaliGemma results.

**Textual Description of the Image** *Does adding a text description of the image greatly improve model performance and confidence?* We initially hypothesized that augmenting the model's input with a textual description of the image would enhance its accuracy and reduce uncertainty, based on the premise that key features necessary for answering the question—already present in the image—would be more accessible to the LLM decoder in text form. However, as illustrated for LLaVA in Figure 6, the results reveal a surprising decrease in answer accuracy and an increase in model uncertainty for configuration I+Q+C$_+$. We observe similar results for PaliGemma (see Appendix I), with minor increases in uncertainty. We argue that these findings indicate the models are already proficient at extracting essential information from the image alone and that the addition of textual information introduces confusion due to redundancy or potential inconsistencies in the image description. Moreover, the high attention allocated to the image description in Figure 6c. underscores the model's sensitivity to textual inputs, which may inadvertently dominate visual cues.

**Prompt Engineering** *Can prompt engineering help VLMs re-balance their attention toward the context?* We modified the initial prompt to direct the model's attention more toward the textual context, which typically receives less emphasis in the standard setting (see Appendix B for implementation details). Given that the complementary context provides information intended to guide the model toward the correct answer, while the contradictory context aims to mislead it, we expected an increase in accuracy for the Q+I+C$_+$ configuration and a decrease for the Q+I+C$_-$ configuration. As shown in Figure 6 a, we observed a decrease in accuracy for Q+I+C$_-$, whereas prompt engineering to emphasize the complementary context resulted in improved answer accuracy. Unexpectedly, these changes are not reflected in the attention attribution: Figure 6c. indicates that prompt engineering does not alter the attention distribution across modalities, as the context does not receive increased attention. This lack of correlation between attention and model performance highlights the necessity for cautious interpretation of attention mechanisms in model predictions (see section 2). Unlike the LLaVA results, PaliGemma exhibited a significant increase in uncertainty (see Appendix I), highlighting the large influence of the image in the standard inputs.

## 6 DISCUSSION & CONCLUSION

**Evaluation of Hypotheses**   Our findings reveal notable insights into the role of each modality in VQA and reasoning tasks. Specifically, we compare all results with our initial hypotheses from subsection 5.1.

*Hypothesis 1 (Including Image):* As expected, introducing the image results in a significant increase in answer accuracy across all VLMs. However, it unexpectedly leads to a decrease in reasoning quality, as in the question-only setting, the models simply acknowledge the absence of the image. We observe a similar pattern in model uncertainty: it decreases for VQ answering but increases in reasoning. As hypothesized, the natural image indeed receives more attention compared to the black baseline image.

*Hypothesis 2 (Including Context):* Consistent with our expectations, the inclusion of complementary context enhances both accuracy and reasoning quality, while contradictory context has a strongly negative effect. However, in VQ answering, the complementary context does not reduce model uncertainty, whereas the contradictory context significantly increases it. In VQA reasoning, contradictory context does not affect uncertainty, and complementary context only slightly decreases it. Generally, the impact of adding context is much stronger in the VQ answering than in the reasoning task. Interestingly, contradictory context can sometimes be beneficial, as it helps to minimize the occurrence of silent failures. Additionally, the models continue to show higher attention toward the image than the context, not supporting our prior hypothesis.

*Hypothesis 3 (Including Image Annotations):* Surprisingly, the image text annotations play a minimal role in enhancing model performance. Although the models exhibit increased attention toward annotated images, the positive impact of these annotations on performance metrics and uncertainty reduction is nearly negligible.

We also investigate methods to guide the model to favor one modality over another to observe the effect on VLM performance. While adding redundant textual information can overwhelm the model and decrease accuracy, prompt engineering can improve predictions without, however, strong changes in attention distribution.

**Limitations and Future Work**   The SI-VQA dataset contains exactly 100 instances. Although relatively small, it is only used for evaluation, not for training or fine-tuning models. Its purpose is to analyze pretrained models' behavior when faced with diverse input configurations. In addition, prior recent research demonstrates that smaller datasets can still yield meaningful results when the quality of instances is high (Polo et al., 2024). Each instance underwent multiple rounds of refinement, ensuring that context descriptions serve only as background information. Despite its reduced size, the dataset exhibits low variance in model performance and uncertainty, as demonstrated in Appendix H, such that adding instances would not alter the conclusions drawn, as they hinge on the modality configurations. The ISI Tool developed in our work ensures that the SI-VQA dataset is extensible and can grow with greater diversity and community input. Our interactive interface also allows people to test new and more advanced modality configurations. In future research, it would be interesting to replicate this study in the reverse scenario—where text is the primary content required for answering, and the image serves as contextual information—to compare whether similar effects of primary and secondary modalities are observed. In future research, it would be interesting to explore tasks where the primary and secondary roles of text and image are reversed, as well as scenarios where both modalities provide distinct yet equally essential information. Investigating how models integrate and prioritize complementary inputs for successful task completion (e.g., answering a question) can offer deeper insights into multimodal information processing. We employ semantic entropy as our unique uncertainty measure, considering other measures for free-form text generation, such as token entropy (Kadavath et al., 2022; Lindley, 1956) or self-expressed uncertainty (Lin et al., 2022; Liao & Wortman Vaughan, 2024), to be significantly less suitable. They either only consider local, token-level uncertainty or rely on the model's potentially biased self-assessment, neither adequately reflecting the overall semantic uncertainty. We validated the expected behavior of semantic entropy across the different configurations using the AUGRC metric in Appendix G.

**Conclusion**   This study is the first to systematically examine the role of contextual information in VQA, evaluating the results based on diverse metrics and distinguishing between answering and reasoning tasks. Leveraging the well-curated SI-VQA dataset and the ISI tool—our interactive, ready-to-use interface—our work aims to provide a deeper understanding of VLM behavior and the influence of each modality. Moreover, we show that our results can also be used to understand and detect model failure in free-form text generation, and setting the stage for future analyses of modality integration across various VLM tasks and the development of VL datasets tailored for this objective.

