## REFERENCES

Jean-Baptiste Alayrac, Jeff Donahue, Pauline Luc, Antoine Miech, Iain Barr, Yana Hasson, Karel Lenc, Arthur Mensch, Katherine Millican, Malcolm Reynolds, et al. Flamingo: a visual language model for few-shot learning. Advances in neural information processing systems, 35:23716–23736, 2022.

Stanislaw Antol, Aishwarya Agrawal, Jiasen Lu, Margaret Mitchell, Dhruv Batra, C Lawrence Zitnick, and Devi Parikh. Vqa: Visual question answering. In Proceedings of the IEEE international conference on computer vision, pp. 2425–2433, 2015.

Emily M Bender, Timnit Gebru, Angelina McMillan-Major, and Shmargaret Shmitchell. On the dangers of stochastic parrots: Can language models be too big?. In Proceedings of the 2021 ACM conference on fairness, accountability, and transparency, pp. 610–623, 2021.

Lucas Beyer, Andreas Steiner, André Susano Pinto, Alexander Kolesnikov, Xiao Wang, Daniel Salz, Maxim Neumann, Ibrahim Alabdulmohsin, Michael Tschannen, Emanuele Bugliarello, Thomas Unterthiner, Daniel Keysers, Skanda Koppula, Fangyu Liu, Adam Grycner, Alexey Gritsenko, Neil Houlsby, Manoj Kumar, Keran Rong, Julian Eisenschlos, Rishabh Kabra, Matthias Bauer, Matko Bošnjak, Xi Chen, Matthias Minderer, Paul Voigtlaender, Ioana Bica, Ivana Balazevic, Joan Puigcerver, Pinelopi Papalampidi, Olivier Henaff, Xi Xiong, Radu Soricut, Jeremiah Harmsen, and Xiaohua Zhai. PaliGemma: A versatile 3B VLM for transfer, July 2024.

Emanuele Bugliarello, Ryan Cotterell, Naoaki Okazaki, and Desmond Elliott. Multimodal pretraining unmasked: A meta-analysis and a unified framework of vision-and-language BERTs. Transactions of the Association for Computational Linguistics, 9:978–994, 2021. doi: 10.1162/tacl_a_00408. URL https://aclanthology.org/2021.tacl-1.58.

Michele Cafagna, Kees van Deemter, and Albert Gatt. What vision-language modelssee'when they see scenes. arXiv preprint arXiv:2109.07301, 2021.

Lin Chen, Jinsong Li, Xiaoyi Dong, Pan Zhang, Yuhang Zang, Zehui Chen, Haodong Duan, Jiaqi Wang, Yu Qiao, Dahua Lin, et al. Are we on the right way for evaluating large vision-language models? arXiv preprint arXiv:2403.20330, 2024.

Michael Dorkenwald, Nimrod Barazani, Cees G. M. Snoek, and Yuki M. Asano. Pin: Positional insert unlocks object localisation abilities in vlms, 2024. URL https://arxiv.org/abs/2402.08657.

Sebastian Farquhar, Jannik Kossen, Lorenz Kuhn, and Yarin Gal. Detecting hallucinations in large language models using semantic entropy. Nature, 630(8017):625–630, June 2024. ISSN 0028-0836, 1476-4687. doi: 10.1038/s41586-024-07421-0.

Stella Frank, Emanuele Bugliarello, and Desmond Elliott. Vision-and-language or vision-for-language? on cross-modal influence in multimodal transformers. arXiv preprint arXiv:2109.04448, 2021.

Itai Gat, Idan Schwartz, and Alex Schwing. Perceptual score: What data modalities does your model perceive? Advances in Neural Information Processing Systems, 34:21630–21643, 2021.

Yash Goyal, Tejas Khot, Douglas Summers-Stay, Dhruv Batra, and Devi Parikh. Making the v in vqa matter: Elevating the role of image understanding in visual question answering. In Proceedings of the IEEE conference on computer vision and pattern recognition, pp. 6904–6913, 2017.

Yangyang Guo, Liqiang Nie, Harry Cheng, Zhiyong Cheng, Mohan Kankanhalli, and Alberto Del Bimbo. On modality bias recognition and reduction. ACM Transactions on Multimedia Computing, Communications and Applications, 19(3):1–22, 2023.

Pengcheng He, Xiaodong Liu, Jianfeng Gao, and Weizhu Chen. Deberta: Decoding-enhanced bert with disentangled attention. In International Conference on Learning Representations, 2021. URL https://openreview.net/forum?id=XPZIaotutsD.

Marcel Hildebrandt, Hang Li, Rajat Koner, Volker Tresp, and Stephan Günnemann. Scene graph reasoning for visual question answering. arXiv preprint arXiv:2007.01072, 2020.

Drew A Hudson and Christopher D Manning. Gqa: A new dataset for real-world visual reasoning and compositional question answering. In Proceedings of the IEEE/CVF conference on computer vision and pattern recognition, pp. 6700–6709, 2019.

Paul F. Jaeger, Carsten T. Lüth, Lukas Klein, and Till J. Bungert. A call to reflect on evaluation practices for failure detection in image classification, 2023. URL https://arxiv.org/abs/2211.15259.

Sarthak Jain and Byron C. Wallace. Attention is not Explanation. In Jill Burstein, Christy Doran, and Thamar Solorio (eds.), Proceedings of the 2019 Conference of the North American Chapter of the Association for Computational Linguistics: Human Language Technologies, Volume 1 (Long and Short Papers), pp. 3543–3556, Minneapolis, Minnesota, June 2019. Association for Computational Linguistics. doi: 10.18653/v1/N19-1357. URL https://aclanthology.org/N19-1357.

Justin Johnson, Bharath Hariharan, Laurens Van Der Maaten, Li Fei-Fei, C Lawrence Zitnick, and Ross Girshick. Clevr: A diagnostic dataset for compositional language and elementary visual reasoning. In Proceedings of the IEEE conference on computer vision and pattern recognition, pp. 2901–2910, 2017.

Gargi Joshi, Rahee Walambe, and Ketan Kotecha. A review on explainability in multimodal deep neural nets. IEEE Access, 9:59800–59821, 2021.

Saurav Kadavath, Tom Conerly, Amanda Askell, Tom Henighan, Dawn Drain, Ethan Perez, Nicholas Schiefer, Zac Hatfield-Dodds, Nova DasSarma, Eli Tran-Johnson, Scott Johnston, Sheer El-Showk, Andy Jones, Nelson Elhage, Tristan Hume, Anna Chen, Yuntao Bai, Sam Bowman, Stanislav Fort, Deep Ganguli, Danny Hernandez, Josh Jacobson, Jackson Kernion, Shauna Kravec, Liane Lovitt, Kamal Ndousse, Catherine Olsson, Sam Ringer, Dario Amodei, Tom Brown, Jack Clark, Nicholas Joseph, Ben Mann, Sam McCandlish, Chris Olah, and Jared Kaplan. Language models (mostly) know what they know, 2022. URL https://arxiv.org/abs/2207.05221.

Ranjay Krishna, Yuke Zhu, Oliver Groth, Justin Johnson, Kenji Hata, Joshua Kravitz, Stephanie Chen, Yannis Kalantidis, Li-Jia Li, David A Shamma, et al. Visual genome: Connecting language and vision using crowdsourced dense image annotations. International journal of computer vision, 123:32–73, 2017.

Bohao Li, Rui Wang, Guangzhi Wang, Yuying Ge, Yixiao Ge, and Ying Shan. Seed-bench: Benchmarking multimodal llms with generative comprehension. arXiv preprint arXiv:2307.16125, 2023.

Victor Weixin Liang, Yuhui Zhang, Yongchan Kwon, Serena Yeung, and James Y Zou. Mind the gap: Understanding the modality gap in multi-modal contrastive representation learning. Advances in Neural Information Processing Systems, 35:17612–17625, 2022.

Q. Vera Liao and Jennifer Wortman Vaughan. AI Transparency in the Age of LLMs: A Human-Centered Research Roadmap. Harvard Data Science Review, (Special Issue 5), may 31 2024. https://hdsr.mitpress.mit.edu/pub/aelql9qy.

Stephanie Lin, Jacob Hilton, and Owain Evans. Teaching models to express their uncertainty in words, 2022. URL https://arxiv.org/abs/2205.14334.

D. V. Lindley. On a Measure of the Information Provided by an Experiment. The Annals of Mathematical Statistics, 27(4):986 – 1005, 1956. doi: 10.1214/aoms/1177728069. URL https://doi.org/10.1214/aoms/1177728069.

Haotian Liu, Chunyuan Li, Qingyang Wu, and Yong Jae Lee. Visual instruction tuning. In NeurIPS, 2023a.

Haotian Liu, Chunyuan Li, Yuheng Li, and Yong Jae Lee. Improved Baselines with Visual Instruction Tuning, May 2024a.

Haotian Liu, Chunyuan Li, Yuheng Li, Bo Li, Yuanhan Zhang, Sheng Shen, and Yong Jae Lee. Llava-next: Improved reasoning, ocr, and world knowledge, January 2024b. URL https://llava-vl.github.io/blog/2024-01-30-llava-next/.

Yuan Liu, Haodong Duan, Yuanhan Zhang, Bo Li, Songyang Zhang, Wangbo Zhao, Yike Yuan, Jiaqi Wang, Conghui He, Ziwei Liu, et al. Mmbench: Is your multi-modal model an all-around player? arXiv preprint arXiv:2307.06281, 2023b.

Pan Lu, Swaroop Mishra, Tony Xia, Liang Qiu, Kai-Wei Chang, Song-Chun Zhu, Oyvind Tafjord, Peter Clark, and Ashwin Kalyan. Learn to explain: Multimodal reasoning via thought chains for science question answering. In The 36th Conference on Neural Information Processing Systems (NeurIPS), 2022.

Omar Moured, Jiaming Zhang, M. Saquib Sarfraz, and Rainer Stiefelhagen. Altchart: Enhancing vlm-based chart summarization through multi-pretext tasks, 2024. URL https://arxiv.org/abs/2405.13580.

OpenAI. GPT-4 Technical Report, March 2024.

Letitia Parcalabescu and Anette Frank. Mm-shap: A performance-agnostic metric for measuring multimodal contributions in vision and language models & tasks. In Proceedings of the 61st Annual Meeting of the Association for Computational Linguistics (Volume 1: Long Papers). Association for Computational Linguistics, 2023. doi: 10.18653/v1/2023.acl-long.223. URL http://dx.doi.org/10.18653/v1/2023.acl-long.223.

Letitia Parcalabescu and Anette Frank. On measuring faithfulness or self-consistency of natural language explanations. In Proceedings of the 62nd Annual Meeting of the Association for Computational Linguistics (Volume 1: Long Papers), pp. 6048–6089, 2024a.

Letitia Parcalabescu and Anette Frank. Do vision & language decoders use images and text equally? how self-consistent are their explanations? arXiv preprint arXiv:2404.18624, 2024b.

Letitia Parcalabescu, Michele Cafagna, Lilitta Muradjan, Anette Frank, Iacer Calixto, and Albert Gatt. VALSE: A task-independent benchmark for vision and language models centered on linguistic phenomena. In Smaranda Muresan, Preslav Nakov, and Aline Villavicencio (eds.), Proceedings of the 60th Annual Meeting of the Association for Computational Linguistics (Volume 1: Long Papers), pp. 8253–8280, Dublin, Ireland, May 2022. Association for Computational Linguistics. doi: 10.18653/v1/2022.acl-long.567. URL https://aclanthology.org/2022.acl-long.567.

Dong Huk Park, Lisa Anne Hendricks, Zeynep Akata, Anna Rohrbach, Bernt Schiele, Trevor Darrell, and Marcus Rohrbach. Multimodal explanations: Justifying decisions and pointing to the evidence. In Proceedings of the IEEE conference on computer vision and pattern recognition, pp. 8779–8788, 2018.

Felipe Maia Polo, Lucas Weber, Leshem Choshen, Yuekai Sun, Gongjun Xu, and Mikhail Yurochkin. tinybenchmarks: evaluating llms with fewer examples. arXiv preprint arXiv:2402.14992, 2024.

Alec Radford, Jong Wook Kim, Chris Hallacy, Aditya Ramesh, Gabriel Goh, Sandhini Agarwal, Girish Sastry, Amanda Askell, Pamela Mishkin, Jack Clark, Gretchen Krueger, and Ilya Sutskever. Learning transferable visual models from natural language supervision, 2021. URL https://arxiv.org/abs/2103.00020.

Nikolaos Rodis, Christos Sardianos, Georgios Th Papadopoulos, Panagiotis Radoglou-Grammatikis, Panagiotis Sarigiannidis, and Iraklis Varlamis. Multimodal explainable artificial intelligence: A comprehensive review of methodological advances and future research directions. arXiv preprint arXiv:2306.05731, 2023.

Sofia Serrano and Noah A. Smith. Is attention interpretable? In Anna Korhonen, David Traum, and Lluís Màrquez (eds.), Proceedings of the 57th Annual Meeting of the Association for Computational Linguistics, pp. 2931–2951, Florence, Italy, July 2019. Association for Computational Linguistics. doi: 10.18653/v1/P19-1282. URL https://aclanthology.org/P19-1282.

Ravi Shekhar, Ece Takmaz, Raquel Fernández, and Raffaella Bernardi. Evaluating the representational hub of language and vision models. In Simon Dobnik, Stergios Chatzikyriakidis, and Vera Demberg (eds.), Proceedings of the 13th International Conference on Computational Semantics - Long Papers, pp. 211–222, Gothenburg, Sweden, May 2019. Association for Computational Linguistics. doi: 10.18653/v1/W19-0418. URL https://aclanthology.org/W19-0418.

Aman Singh Thakur, Kartik Choudhary, Venkat Srinik Ramayapally, Sankaran Vaidyanathan, and Dieuwke Hupkes. Judging the judges: Evaluating alignment and vulnerabilities in llms-as-judges, 2024. URL https://arxiv.org/abs/2406.12624.

Jeremias Traub, Till J. Bungert, Carsten T. Lüth, Michael Baumgartner, Klaus H. Maier-Hein, Lena Maier-Hein, and Paul F Jaeger. Overcoming common flaws in the evaluation of selective classification systems, 2024. URL https://arxiv.org/abs/2407.01032.

Sarah Wiegreffe and Yuval Pinter. Attention is not not explanation. In Kentaro Inui, Jing Jiang, Vincent Ng, and Xiaojun Wan (eds.), Proceedings of the 2019 Conference on Empirical Methods in Natural Language Processing and the 9th International Joint Conference on Natural Language Processing (EMNLP-IJCNLP), pp. 11–20, Hong Kong, China, November 2019. Association for Computational Linguistics. doi: 10.18653/v1/D19-1002. URL https://aclanthology.org/D19-1002.

Thomas Wolf, Lysandre Debut, Victor Sanh, Julien Chaumond, Clement Delangue, Anthony Moi, Pierric Cistac, Tim Rault, Rémi Louf, Morgan Funtowicz, Joe Davison, Sam Shleifer, Patrick von Platen, Clara Ma, Yacine Jernite, Julien Plu, Canwen Xu, Teven Le Scao, Sylvain Gugger, Mariama Drame, Quentin Lhoest, and Alexander M. Rush. Huggingface's transformers: State-of-the-art natural language processing, 2020. URL `https://arxiv.org/abs/1910.03771`.

Xiang Yue, Yuansheng Ni, Kai Zhang, Tianyu Zheng, Ruoqi Liu, Ge Zhang, Samuel Stevens, Dongfu Jiang, Weiming Ren, Yuxuan Sun, Cong Wei, Botao Yu, Ruibin Yuan, Renliang Sun, Ming Yin, Boyuan Zheng, Zhenzhu Yang, Yibo Liu, Wenhao Huang, Huan Sun, Yu Su, and Wenhu Chen. Mmmu: A massive multi-discipline multimodal understanding and reasoning benchmark for expert agi. In Proceedings of CVPR, 2024a.

Xiang Yue, Yuansheng Ni, Kai Zhang, Tianyu Zheng, Ruoqi Liu, Ge Zhang, Samuel Stevens, Dongfu Jiang, Weiming Ren, Yuxuan Sun, Cong Wei, Botao Yu, Ruibin Yuan, Renliang Sun, Ming Yin, Boyuan Zheng, Zhenzhu Yang, Yibo Liu, Wenhao Huang, Huan Sun, Yu Su, and Wenhu Chen. Mmmu: A massive multi-discipline multimodal understanding and reasoning benchmark for expert agi. In Proceedings of CVPR, 2024b.

Rowan Zellers, Yonatan Bisk, Ali Farhadi, and Yejin Choi. From recognition to cognition: Visual commonsense reasoning. In Proceedings of the IEEE/CVF conference on computer vision and pattern recognition, pp. 6720–6731, 2019.

Lianmin Zheng, Wei-Lin Chiang, Ying Sheng, Siyuan Zhuang, Zhanghao Wu, Yonghao Zhuang, Zi Lin, Zhuohan Li, Dacheng Li, Eric P. Xing, Hao Zhang, Joseph E. Gonzalez, and Ion Stoica. Judging llm-as-a-judge with mt-bench and chatbot arena. In Proceedings of the 37th International Conference on Neural Information Processing Systems, NIPS '23, Red Hook, NY, USA, 2024. Curran Associates Inc.

Yuke Zhu, Oliver Groth, Michael Bernstein, and Li Fei-Fei. Visual7w: Grounded question answering in images. In Proceedings of the IEEE conference on computer vision and pattern recognition, pp. 4995–5004, 2016.

## A  MODALITY ATTENTION AGGREGATION

From the VLMs, we extract an attention matrix $A_t \in \mathbb{R}^{t \times t}$ for each output token $t$, where the matrix size grows with the number of predicted tokens. Each row represents a token as a query, and each column corresponds to a token as a key. Therefore, each row $r$ contains the attention coefficients of each token $i$ with respect to the token $r$: $A[r, i] = \alpha_{r,i} = q_r^\top k_i$. For each token $t$, we focus only on the attention of preceding tokens, represented by $v_t^\top = A[-1] \in \mathbb{R}^{1 \cdot t}$. $v_t$ is normalized so that the attention coefficients of all preceding tokens to $t$ sum to 1.

This process is repeated for all output tokens $t \in [1, T]$ where $T$ is the output length, yielding all normalized attention vectors $v_t, t \in [1, T]$. These are averaged to produce the final attention vector for the VLM output—$V_A \in \mathbb{R}^{N_0}$ for answers and $V_R \in \mathbb{R}^{N_0 + T_A + N_1}$ for reasoning. Here, $N_0$ is the size of the input prompt including image tokens, question, and eventually context, $T_A$ is the answer length, and $N_1$ is the size of the second prompt asking for an explanation.

To calculate the attention given to different modalities, we sum the attention coefficients based on the token positions in the averaged attention vector $V_A$ or $V_R$, resulting in normalized relevance scores: $R_I$ for the image, $R_Q$ for the question, and $R_C$ for the context, which sum to 1, i.e., $R_I + R_Q + R_C = 1$. This is done by adding hooks into the LLaVA architecture to capture the start and end positions of the question, image, and context tokens. Due to dynamic high resolution, the number of image tokens can vary significantly even with minimal image perturbations.

---

**Algorithm 1** Attention Attribution Computation

1: **Input:** Attention matrix $A_t \in \mathbb{R}^{t \times t}$ for each token $t$, Token positions for Image ($I$), Question ($Q$), and Context ($C$), Length of output $T$
2: **Output:** Normalized relevance scores $R_I, R_Q, R_C$
3: **for** $t \in [1, T]$ **do**
4:  Extract attention for preceding tokens: $v_t^\top = A[-1] \in \mathbb{R}^{1 \times t}$
5:  Normalize attention coefficients: $v_t \leftarrow \frac{v_t}{\sum_{i=1}^t v_t[i]}$
6: **end for**
7: Compute average attention vector: $V = \frac{1}{T} \sum_{t=1}^T v_t^T$
8: Compute total attention for each modality:
9: $R_I \leftarrow \sum_{i \in I} V[i]$
10: $R_Q \leftarrow \sum_{i \in Q} V[i]$
11: $R_C \leftarrow \sum_{i \in C} V[i]$
12: Normalize relevance scores:
13: $R_I \leftarrow \frac{R_I}{R_I + R_Q + R_C}$
14: $R_Q \leftarrow \frac{R_Q}{R_I + R_Q + R_C}$
15: $R_C \leftarrow \frac{R_C}{R_I + R_Q + R_C}$
16: **Return:** $R_I, R_Q, R_C$

---

## B  HYPERPARAMETER AND EVALUATION PROMPTS

In reasoning quality evaluation, GPT-4o is prompted to rate the reasoning from 0 to 10, using the prompt: "Rate the explanation's quality from 0 to 10. Give 10 for detailed, well-argued, and correct explanations. Give 0 for a poorly reasoned, wrong, or single-word explanation based on the question and image. Don't rate too harshly, use the full scale and output only the final score". During uncertainty computation, the number of the sampled outputs and the sampling temperature $T$ are set to 10 and 0.9 respectively. We use conditional probabilistic sampling. See Appendix A for the attention attribution implementation.

In the prompt engineering ablation study in subsection 5.5 we use the prompt "Answer the question only with Yes or No. Answer based on the context provided in the text." for the answer, followed by "Explain your answer based on the context provided in the text:" for reasoning.

## C  SI-VQA DATASET BASELINE

In the baseline configuration of the SI-VQA Dataset, the image tokens carry no meaningful information. We experiment with several methods to remove image data: (1) replacing the image with black pixels, (2) adding random

Gaussian noise centered on (0,0,0), or (3) omitting the image tokens entirely. In (1) and (2), the image size is by default set to $256 \times 256$ pixels. Across our experiments, we observed similar results when using black pixels or noise. However, when no image input is provided, the model produces incoherent predictions and reasoning, often generating "\n". Scenario (3) is out-of-distribution for any VLMs, leading to nonsensical outputs, as the model was trained exclusively in a multimodal context and has never encountered unimodal (text-only) input.

---

### (1) Baseline configuration with black image

**Question:** Is there a yellow and black frog in this school book?
**Ground truth answer:** Yes

---

**Answer:** No
**Reasoning:** The image you provided is completely black, which does not allow for the identification of any objects or creatures, including a yellow and black frog.

---

### (2) Baseline configuration with noisy image

**Question:** Is there a yellow and black frog in this school book?
**Ground truth answer:** Yes

---

**Answer:** No
**Reasoning:** The image is a close-up of a colorful, pixelated background. There is no frog visible in the image.

---

### (3) Baseline configuration with no image token

**Question:** Is there a yellow and black frog in this school book?
**Ground truth answer:** Yes

---

**Answer:** 是 이 이 \n \n Is a \n\n
**Reasoning:**         10

# D  SI-VQA BENCHMARK EXAMPLES

Each instance of the SI-VQA Dataset was double-checked and validated by three individuals.

---

**Boiling.jpg**

**Question:** Is the water hot?
**Ground truth answer:** Yes
**LLaVA 7B:** No. The water is not hot.
**LLaVA-Next 7B:** No. The image you provided is too dark and lacks any discernible features to determine if the water is hot or not.

---

**Question:** Is the water hot? 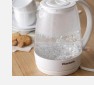
**LLaVA 7B:** No. The water in the pitcher is cold.
**LLaVA-Next 7B:** No. The image shows a pitcher of water with ice cubes, which suggests that the water is cold.

---

**Question:** Is the water hot? 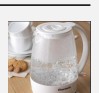
**Complementary Context:** Kettles, essential for boiling water, are commonly used for making tea, coffee, or instant meals, and come in various designs, including traditional stove top and modern electric models.
**LLaVA 7B:** Yes. The water in the kettle is hot.
**LLaVA-Next 7B:** No. The image shows a water kettle with water inside it, but it is not hot. The water appears to be at room temperature or slightly cooler, as indicated by the condensation on the outside of the kettle.

---

**Question:** Is the water hot? 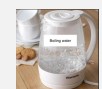
**Contradictory Context:** Glass pitchers are often used to serve ice-cold beer, making them a popular choice in bars and restaurants for sharing drinks in a casual, communal setting.
**LLaVA 7B:** No. The water in the pitcher is not hot. It is ice-cold, which is a common feature of glass pitchers used for serving beer.
**LLaVA-Next 7B:** No. The image shows a glass pitcher filled with ice and water, which is not hot. The ice is used to keep the water cold, and the pitcher is designed to serve multiple people at once. This type of pitcher is commonly used for serving cold beverages, such as water or iced tea, in a social setting.

---

**Question:** Is the water hot? 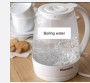
**LLaVA 7B:** No. Boiling water is water that is heated to the point of boiling, which is the temperature at which water vaporizes. In this image, the water in the kettle is not hot, as it is described as "boiling water" and "cold water."
**LLaVA-Next 7B:** Yes. The image shows a water pitcher with the label "Boiling water" on it, which suggests that the water inside is being heated to boiling point. The presence of steam and the fact that the water is being heated in a pitcher designed for boiling water further supports the idea that the water is hot.

---

**Question:** Is the water hot? 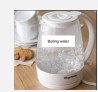
**Complementary Context:** Kettles, essential for boiling water, are commonly used for making tea, coffee, or instant meals, and come in various designs, including traditional stove top and modern electric models.
**LLaVA 7B:** Yes. The water in the kettle is hot, as indicated by the steam rising from it.
**LLaVA-Next 7B:** No. The image shows a water kettle with a label that reads "Boiling water." However, the water in the kettle appears to be cold, as indicated by the condensation on the outside of the kettle. This suggests that the water has not yet reached its boiling point.

---

**Question:** Is the water hot?
**Contradictory Context:** Glass pitchers are often used to serve ice-cold beer, making them a popular choice in bars and restaurants for sharing drinks in a casual, communal setting.
**LLaVA 7B:** No. The water in the pitcher is not hot. It is ice-cold, which is a common feature of glass pitchers used for serving beer.
**LLaVA-Next 7B:** No. The image shows a water pitcher with the label "Boiling water" on it, but the water inside appears to be cold. The context of the image, which includes a wooden table and a few cookies, suggests that the water is being used for a hot beverage, such as tea or coffee. However, the water itself is not boiling at the moment the image was taken.

## Painting_asia.jpg

**Question:** Is the painting of european descent?

**Ground truth answer:** No

**LLaVA 7B:** No. The painting is of African descent.

**LLaVA-Next 7B:** No. The image you provided is too dark and lacks detail to determine if it is a painting of European descent. The image is not clear enough to make an accurate assessment.

---

**Question:** Is the painting of european descent? 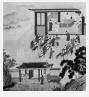

**LLaVA 7B:** No. The painting is of Asian descent.

**LLaVA-Next 7B:** No. The painting is not of European descent. It appears to be a traditional East Asian artwork, likely depicting a scene from a historical period in China or Japan, given the style of clothing and the architectural elements.

---

**Question:** Is the painting of european descent? 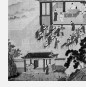

**Complementary Context:** Scholars and students engaged in study and teaching within an ancient Chinese educational setting. The intricate details of the architecture and the serene outdoor environment emphasize the importance of learning and intellectual pursuit in classical Chinese culture.

**LLaVA 7B:** Yes. The painting is of an ancient Chinese educational setting, featuring scholars and students engaged in study and teaching. The intricate details of the architecture and the serene outdoor environment emphasize the importance of learning and intellectual pursuit in classical Chinese culture.

**LLaVA-Next 7B:** No. The painting is not of European descent. It depicts a traditional Chinese scene, with scholars and students engaged in study and teaching within an ancient Chinese educational setting. The architecture and the serene outdoor environment are characteristic of Chinese art and culture, and the painting is not of European origin.

---

**Question:** Is the painting of european descent? 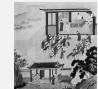

**Contradictory Context:** In an old German schoolhouse, students diligently engage in their lessons, surrounded by the timeless charm of wooden desks, slate chalkboards, and the warm glow of afternoon sunlight filtering through the tall, arched windows.

**LLaVA 7B:** Yes. Yes.

**LLaVA-Next 7B:** Yes. The painting depicts a scene that is characteristic of traditional European schools, with students seated at wooden desks, using slate chalkboards, and the presence of tall, arched windows suggests a European architectural style. The clothing of the students and the overall composition of the painting are consistent with European art and culture.

---

**Question:** Is the painting of european descent? 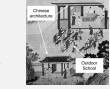

**LLaVA 7B:** No. The painting is of Chinese descent.

**LLaVA-Next 7B:** No. The painting is not of European descent. It depicts a traditional Chinese scene, with Chinese architecture, clothing, and people engaged in activities such as writing and reading. The style of the painting, including the clothing and the architectural elements, is indicative of a Chinese setting, which suggests that the painting is not of European descent.

---

**Question:** Is the painting of european descent? 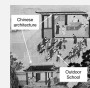

**Complementary Context:** Scholars and students engaged in study and teaching within an ancient Chinese educational setting. The intricate details of the architecture and the serene outdoor environment emphasize the importance of learning and intellectual pursuit in classical Chinese culture.

**LLaVA 7B:** No. The painting is of Chinese descent, depicting a scene of scholars and students engaged in study and teaching within an ancient Chinese educational setting. The intricate details of the architecture and the serene outdoor environment emphasize the importance of learning and intellectual pursuit in classical Chinese culture.

**LLaVA-Next 7B:** No. The painting is not of European descent. It depicts a traditional Chinese scene, with Chinese architecture, clothing, and hairstyles, which are characteristic of East Asian art. The style of the painting, including the brushwork and the subject matter, are indicative of Chinese art and culture.

---

**Question:** Is the painting of european descent? 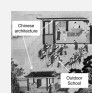

**Contradictory Context:** In an old German schoolhouse, students diligently engage in their lessons, surrounded by the timeless charm of wooden desks, slate chalkboards, and the warm glow of afternoon sunlight filtering through the tall, arched windows.

**LLaVA 7B:** No. The painting is of Chinese descent, depicting a scene from a Chinese school.

**LLaVA-Next 7B:** Yes. The painting depicts a traditional European schoolhouse, with students seated at wooden desks and chalkboards, and the architecture and style of the building suggest a European influence. The presence of tall, arched windows and the overall composition of the scene are characteristic of European art and architecture.

# E ISI TOOL

The interactive tool can be used to analyze VLMs with the provided SI-VQA Dataset and follows a main pipeline that consists of three main steps: 1) Data & Model Selection 2) Interventions on Image, Context, and Question, and 3) Evaluation. Figure 7 gives an overview of this pipeline.

## E.1 GENERAL INFORMATION

**Users** The application is catered toward researchers, developers, and other users with a basic understanding of VLMs, who are interested in interpreting model behavior through semantic interventions on VLMs. By enabling fast-paced iterations in a human-in-the-loop scenario, it allows the building of intuitions before scaling experiments in large-scale projects.

**System Requirements** ISI for VLMs is an interactive tool embedded in a locally hosted web application requiring a computer with sufficient VRAM for VLM inference. The minimum required VRAM for a 4bit-quantized LLAVA 7B model is around 8GB while LLaVA-Vicuna and LLaVA-Next require 12GB. The computation of the semantic entropy with the DeBERTa model requires an additional 7GB. The exact amount of VRAM depends on the amount of input tokens.

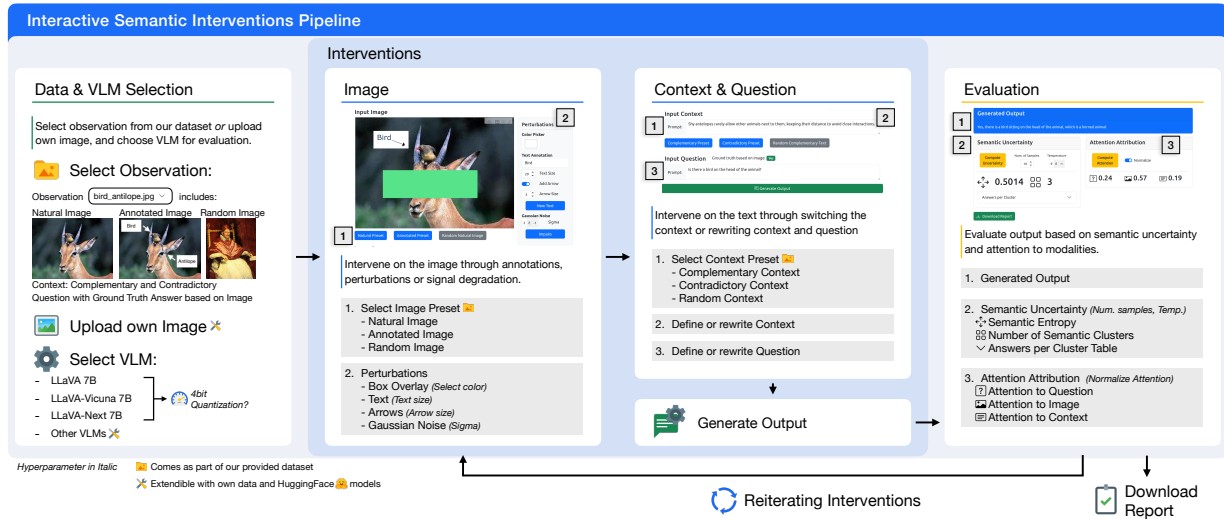

Figure 7: Illustration of the evaluation pipeline used in the ISI for VLMs to enable interactive exploration of VLM behavior under various scenarios. It consists of three main stages: 1) Data & VLM Selection: Users choose an observation either from the provided SI-VQA Dataset or upload their own, and select a VLM for evaluation. 2) Interventions on Image, Context & Question: The selected image can be altered through presets or perturbations, and the context or question can be edited or also switched with presets. 3) Evaluation: The output is analyzed for semantic uncertainty and attention attribution, allowing for iterative refinement of interventions.

## E.2 INTERACTIVE SEMANTIC INTERVENTION PIPELINE

**Data & Model Selection:** As the first step, a user either chooses an observation from the SI-VQA Dataset or uploads their custom image. Each observation from the dataset includes an image, corresponding context, and a question with a ground truth answer, as well as the presets for the annotated images and contradictory and complementary context. The corresponding image, context, and question are displayed. In the next step, the user selects a VLM (LLaVA, LLaVA-Vicuna, LLaVA-Next) and the number of parameters (7B, 11B, 32B) in two separate drop-down menus. 4-bit quantization can be enabled to reduce the computational load and VRAM requirements.

**Interventions on the Image:** For interventions on the image, ISI allows the user two main functionalities. First, on the proposed SI-VQA Dataset the user can select for each observation three different image presets (natural image without modifications, annotated image with hand-crafted annotations, and random natural image from the dataset) by selecting the respective buttons. Second, ISI allows perturbing the image directly in the tool by overlaying boxes

with selectable colors, inserting and modifying text, adding directional arrows, and introducing Gaussian noise with adjustable noise values.

**Interventions on Context & Question:** To facilitate the user's ability to observe how various contexts and questions affect the model's performance two functionalities are supported. In the proposed SI-VQA Dataset, users can choose from three distinct context presets—complementary, contradictory, or random—by selecting the respective button, automatically updating the content in the text input fields. Additionally, the user can manually edit the context and question in these fields.

**Evaluation** The evaluation is designed to enable quantitative analysis of how interventions on image and text impact the behavior of the selected VLM. At the top, the current input is visualized to always relate the evaluation results to the correct input. Below, the generated output, semantic uncertainty, and attention attribution are shown. For computational reasons, each evaluation can be started separately.

The semantic uncertainty tab allows users to evaluate model uncertainty by clustering sampled outputs according to their semantic meaning. This feature highlights the range of semantic differences in the outputs and calculates semantic entropy, providing a comprehensive view of the model's overall uncertainty. It displays key metrics, such as semantic entropy and the number of semantic clusters, which are influenced by adaptable hyperparameters like the number of samples and the sampling temperature. For deeper exploration, the "Answers per Cluster" dropdown provides a table displaying all sampled answers along with their assigned semantic clusters. This table enables users to examine the full range of generated outputs and understand the semantic similarities within each cluster. To evaluate the significance of each of the three inputs during generation, the attention attribution tab displays the absolute or relative attention assigned to the question, context, and image input tokens.

To provide a contextual understanding of the current observation, the tool additionally displays average values for attention attribution and semantic entropy across the entire SI-VQA Dataset based on each VLM architecture. These averages are shown when hovering over the relevant values.

**Export** The results of one iteration can be exported as a PDF to facilitate the systematic collection of example cases for further analysis and to support the transition from initial qualitative insights to small-scale quantitative evaluation. After the analysis, users can download a comprehensive report that includes the image, context, question, detailed model setup, hyperparameters, and all computed evaluation metrics.

# F  4BIT QUANTIZATION

**a.**  Difference 4bit/32bit Quantization VQ Answering Accuracy

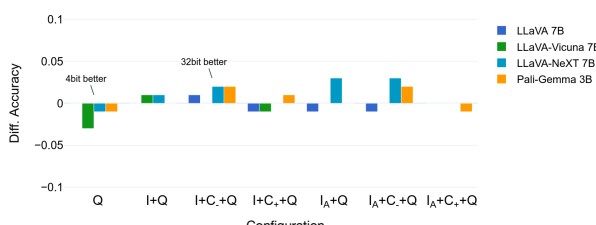

**b.**  Difference 4bit/32bit Quantization Sem. Entropy Answering

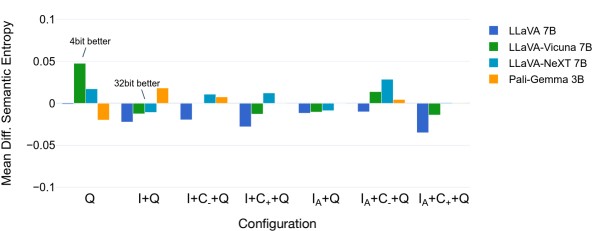

**c.**  Difference 4bit/32bit Quantization Sem. Entropy Reasoning

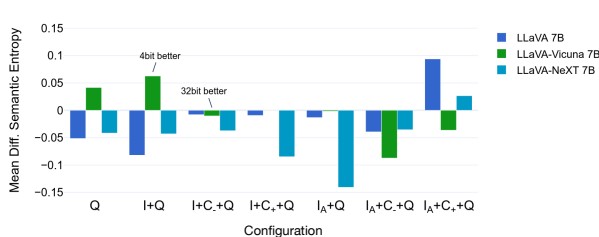

**d.**  Difference 4bit/32bit Quantization Attention Answering

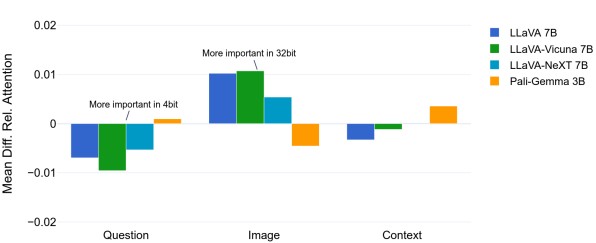

**e.**  Difference 4bit/32bit Quantization Attention Reasoning

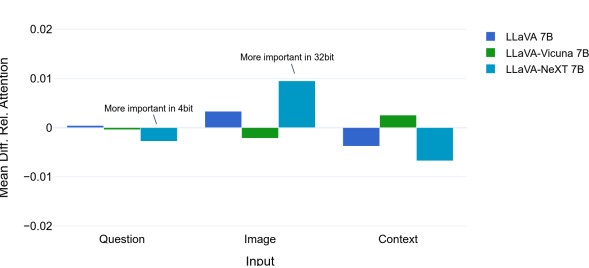

Figure 8: Difference in performance between the 32Bit and 4Bit quantized versions of the models for a. VQA accuracy, b. VQA semantic entropy, c. reasoning semantic entropy, d. VQA attention attribution, and e. reasoning attention attribution.

To quantify the effect of different model sizes, we additionally perform all experiments also with the same models but 4Bit quantized, as there are no, e.g., 3B parameter LLaVA versions. Figure 8 shows the difference in results for all five experiments between the 32Bit and 4Bit models. To our surprise, the difference in accuracy is not that large. Results for the question-only configuration are not meaningful as both models randomly guess. However, in terms of model uncertainty, the 32Bit model usually scores better. Mean differences in attention distribution are almost neglectable as they are at a maximum of 0.01 percentage points. The results show that for simple VQA, quantized models can achieve almost similar accuracy than their significantly larger 32Bit counterparts.

# G  VALIDATING SEMANTIC ENTROPY

Any assumptions made on the uncertainty of the models should be reflected in the AUGRC. We observe in Figure 4 that in the case of contradicting image and context (Q+I+C$_-$) the AUGRC goes down for all models, reducing the overconfidence in wrong classified samples, which is correctly captured by the semantic entropy. As the model makes more mistakes in configuration Q+I+C$_-$, the set size of wrong classified samples is larger. In the case of Q+I or Q+I+C$_+$ the AUGRC rises again as there is no confidence reducing context. Additionally, the accuracy is higher in the case of Q+I+C$_+$, reducing the set size of wrong classified samples. This empirical evaluation shows we can use the AUGRC to quantify and evaluate the performance of semantic entropy for model failure detection. To our knowledge, this is the first time semantic entropy has been evaluated for failure detection.

# H   ADDITIONAL RESULTS

## H.1   VQA ACCURACY AND QUALITY

**a.** VQA Confusion Matricies

Figure 9: Confusion matrices and accuracy values for all model architectures and configurations.

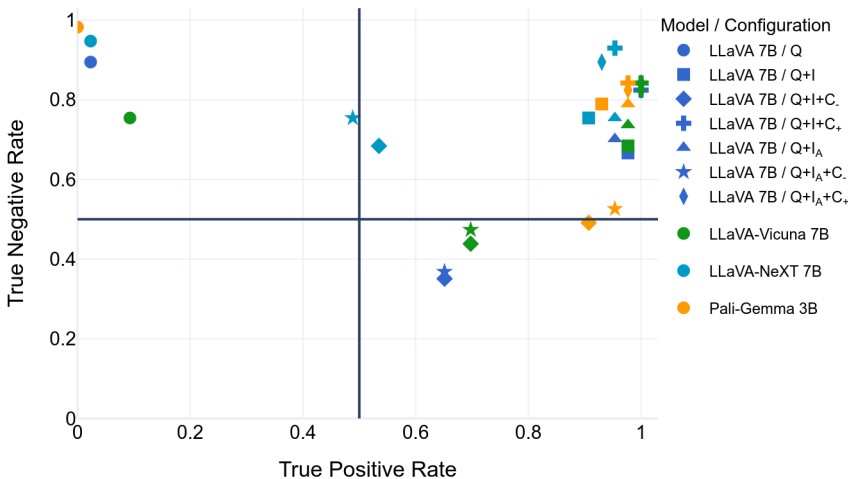

Figure 10: True Negative Rate versus the True Positive Rate in the VQA task of all model architectures and configurations. Values below 0.5 on one axis indicate worse than random guessing performance.

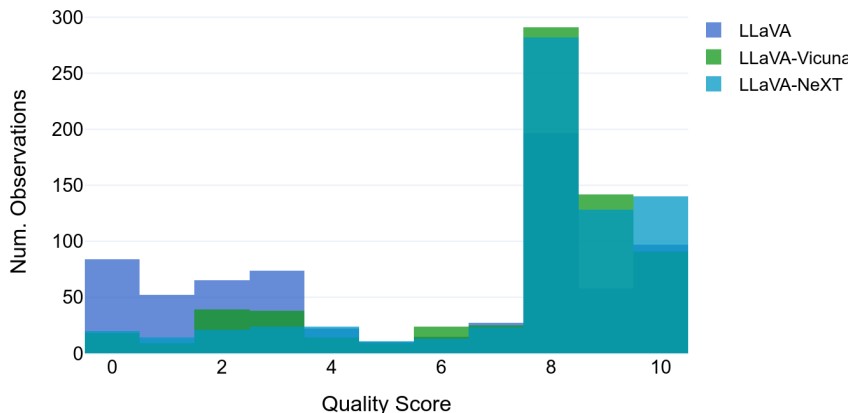

Figure 11: Distribution of reasoning quality scores by GPT-4o for all LLaVA architectures. We observe lower values for the standard LLaVA and a high bias towards the quality score of "8" for all models.

## H.2 SEMANTIC ENTROPY

**a.** Semantic Entropy for VQ Answering (lower is better)

**b.** Semantic Entropy for VQ Reasoning (lower is better)

Figure 12: Distribution of semantic entropy for VQ answering and reasoning task for all model architectures and configurations.

## H.3 ATTENTION

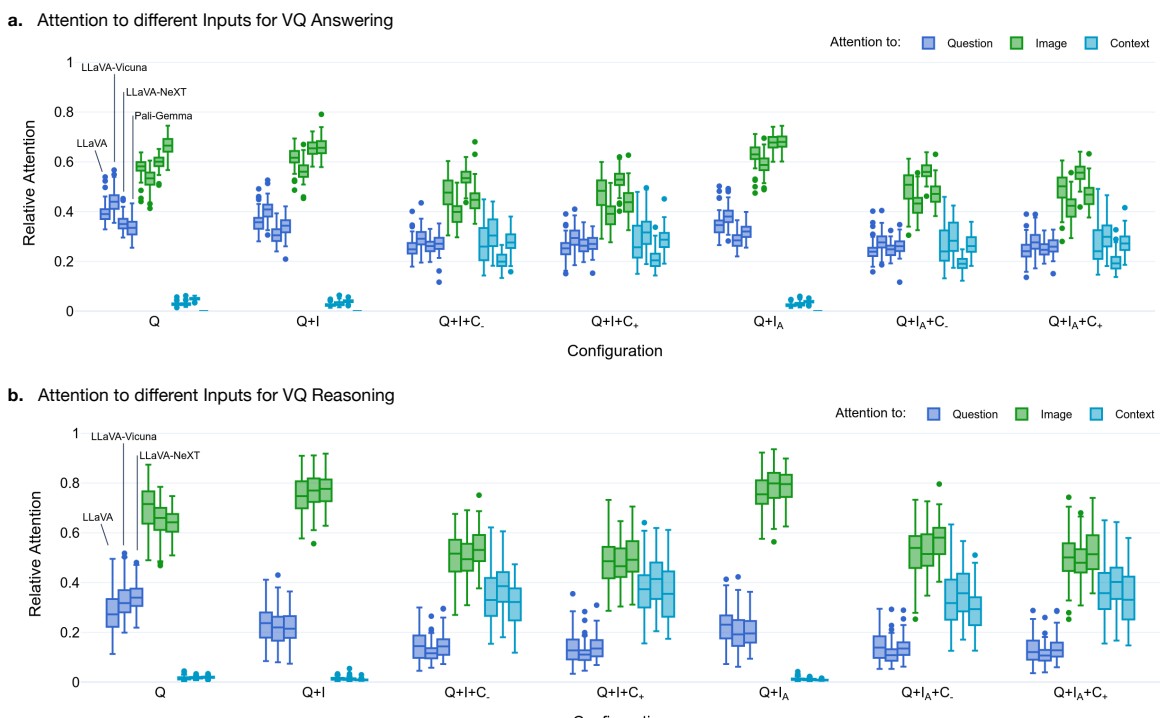

Figure 13: Attention attribution to the question (Q), image (I), and context ($C_{+/-}$) by the three LLaVA models and PaliGemma and the seven modality configurations, during the answering (a.) and the reasoning (b.) in the VQA tasks. Variance across samples is greater for the reasoning than the answering process.

## H.4 PEARSON CORRELATION BETWEEN ATTENTION AND ACCURACY

Table 1: The Pearson correlation coefficients (PCC) between the attention to the different inputs and the accuracy per sample. Correlation is between -0.07 and 0.9 for all inputs and model architectures and can fluctuate between architectures.

| Model Architecture: | LLAVA 7B | LLAVA-Vicuna 7B | LLAVA-NeXT 7B | PaliGemma 3B |
|---|---|---|---|---|
| Question | -0.034 | -0.034 | -0.074 | -0.044 |
| Image | 0.087 | 0.002 | -0.005 | -0.014 |
| Context | -0.036 | 0.016 | 0.045 | 0.026 |

# I  REBALANCING MODALITY IMPORTANCE FOR PALIGEMMA 3B

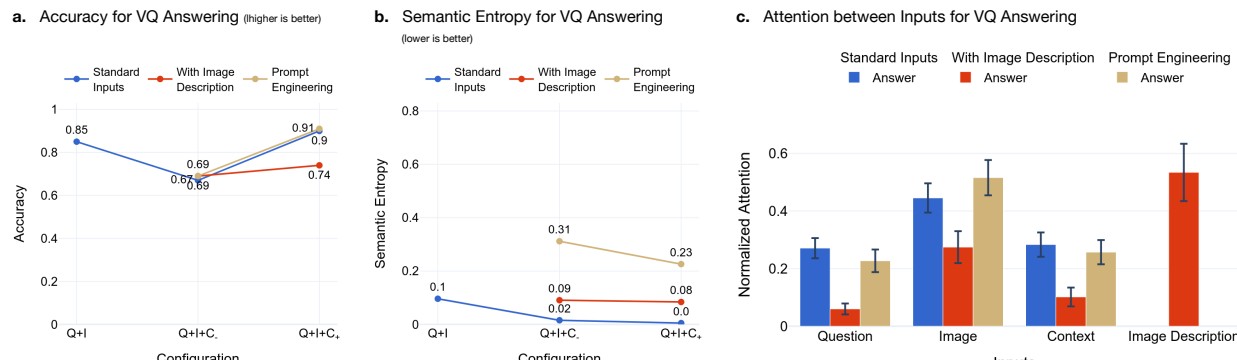

Figure 14: Two strategies to adjust modality importance: incorporating the image's textual description into the input and modifying the prompt to shift more attention toward the context. The impact of these changes is evaluated based on a. answer accuracy, b. model uncertainty, and c. attention attribution. The hereby experiments were conducted with the PaliGemma model.