# OpenReview forum: "Why context matters in VQA & Reasoning: Semantic interventions for VLM input modalities"
_ICLR.cc/2025/Conference — Submitted to ICLR 2025_

### Official Review · Reviewer_9N8L · 2024-10-21

**Soundness:** 1
**Presentation:** 2
**Contribution:** 1
**Rating:** 5
**Confidence:** 5

**Summary:**

This paper presents to evaluate the robustness of VLLMs.
In particular, there are two dimensions that the advocated evaluation protocol considers:
1) modality bias - whether VLLMs make predictions based on the linguistic relations;
2) context - whether the context helps in reasoning.

Based on this idea, this paper collects a new dataset and then evaluates various VLLMs, including LLaVA, and Pali-Gemma.
Besides, the authors also provide some analysis from the dimension of semantic entropy and attention distribution.

**Strengths:**

- The studied problem - the robustness of VLLMs, is practical and interesting for the research community.
- The authors adopt two different families of models for evaluation, including both LLaVA and Pali-Gemma.
- There are some more dimensions that are considered by this paper, like semantic entropy and attention distribution.

**Weaknesses:**

- The biggest limitation of this paper lies in its limited dataset size.
Specifically, there are only 100 instances of the collected dataset.
From this point of view, most of the conclusions from this work might be plausible and not stand.
Additionally, we cannot name a scale of such a dataset as ``comprehensive``.
- The authors are suggested to test larger model sizes, such as 13B models - LLaVA-1.5-vicuna-13B.
- It seems like there is a strong connection between this work and several well-studied problems such as modality bias (language prior) in VQA [1][2], and visual commonsense reasoning (VCR) [3].

[1] On Modality Bias Recognition and Reduction.

[2] Mind the Gap: Understanding the Modality Gap in Multi-modal Contrastive Representation Learning.

[3] From Recognition to Cognition: Visual Commonsense Reasoning.

**Questions:**

See the weakness part for a detailed explanation.

---

> ### Author Response · Authors · 2024-11-21
> **Rebuttal to Reviewer 9N8L - PART 1/2**
>
> Thank you for your comments and for raising these important points. We appreciate the opportunity to clarify our findings and address your concerns.
>
> 1. Dataset size.
>
> Regarding the dataset size,
>
> - **Evaluation purpose.** We emphasize that our dataset is only used for evaluation, not for training or fine-tuning models. Its purpose is to analyze pretrained models’ behavior when faced with diverse input configurations.
> - **TinyBenchmark.** While the SI-VQA dataset we designed remains small, prior recent research [1] demonstrates that smaller datasets can still yield meaningful results when the quality of instances is high. Each instance of the SI-VQA Dataset underwent multiple rounds of refinement, ensuring that context descriptions serve only as background information without providing enough detail to answer the question independently.
> - **Low Variance.** Despite the small size, the dataset exhibits low variance in model performance and uncertainty, as demonstrated in the Appendix H. Increasing the number of instances would not alter the conclusions drawn, as they hinge on the modality configurations rather than the dataset size. Instead, rebalancing the role of modalities, such as prioritizing text and introducing spurious visual information, would yield new insights—a direction we are actively exploring.
> - **Scalability and Community Engagement.** To ensure data quality and consistency, we intentionally limited the dataset size to 100 well-curated instances. However, we have also developed a ready-to-use interactive interface that allows the community to generate additional instances with new images and textual descriptions. This ensures that our SI-VQA dataset is extensible and can grow with greater diversity and community input. See Appendix E for details about the Interface and instructions to create new data instances and the link to the code to try it out: https://gitlab.com/dekfsx1/isi-vlm.
>
> [1] tinyBenchmarks: evaluating LLMs with fewer examples
>
> *In the revised version of the paper, we integrated this argumentation in the Limitations and Future Directions paragraph in the Discussion.*
>
> 2. The authors are suggested to test larger model sizes, such as 13B models - LLaVA-1.5-vicuna-13B.
>
> We appreciate the reviewer’s suggestion to test larger model sizes, such as 13B, but our study is focused on observing performance changes across different strategies rather than achieving the highest possible accuracy. Our chosen 7B model aligns with the difficulty of our VQA dataset and provides an optimal balance for meaningful performance variation. In addition, our ablation study in Appendix Section F shows minimal information loss with quantization, supporting the suitability of the 7B model size for our analysis. Using a larger model would obscure these nuanced differences and shift focus away from our core objective, thus we only recommended it for further research.
>
> *We have added a sentence in section 4.1: “We restrict our analysis to 7B models because this number of parameters aligns with the difficulty of our VQA dataset and provides an optimal balance for meaningful performance variation.”*

---

> > ### Comment · Reviewer_9N8L · 2024-11-26
> > **Response to Authors' Rebuttal**
> >
> > I thank the authors for their extensive response to my concerns.
> > The authors partially addressed my concerns about the relationship to other similar studies such as the modality gap.
> >
> > But the dataset size is still too limited and using a larger model for evaluation will help readers better understand the reasoning steps.
> > Combining the above ideas, I believe the paper, with a modest revision, will make a solid contribution in the future. Therefore, I will increase my score to 5 and hope the authors can improve their work.

---

> ### Author Response · Authors · 2024-11-21
> **Rebuttal to Reviewer 9N8L - PART 2/2**
>
> 3. It seems like there is a strong connection between this work and several well-studied problems such as modality bias (language prior) in VQA [1][2], and visual commonsense reasoning (VCR) [3].
>
>
> We thank the reviewer for bringing these three research papers to our attention. While these works focus on the role of modalities in vision-language models, their direct relevance to our study is limited due to differences in objectives and scope. Before elaborating further, we would like to clarify what our paper is not:
> 1. We are not proposing a new large-scale multimodal benchmark aimed at achieving state-of-the-art results in multimodal learning.
> 2. We are not attempting to resolve modality bias.
>
> With these distinctions in mind, we would like to explain how the cited works relate to our paper:
>
> [1] "On Modality Bias Recognition and Reduction"
>
> This paper explores modality bias in VLMs and proposes a fine-tuning strategy using a de-biasing loss function. While we acknowledge the importance of addressing modality bias, our objective is not to mitigate these biases directly. Instead, we aim to study and understand such biases by experimenting with modality configurations, specifically in the context of the SI-VQA dataset. Using pre-trained VLMs that come with inherent inductive biases, we test two methods to rebalance modality importance (“de-biasing”) and report diverse and, at times, surprising conclusions. Our work complements [1] by exploring the interplay of modalities rather than focusing on reducing modality bias.
>
> [2] "Mind the Gap: Understanding the Modality Gap in Multi-modal Contrastive Representation Learning"
>
> This paper investigates the existence of a modality gap in the embedding space under various initializations and linear activations, offering an analysis of how the gap impacts downstream performance and fairness. While we find this work highly relevant to understanding modality roles, our focus is different: we examine the effects of modality interplay on VLM behavior (e.g., performance, uncertainty) and characteristics (e.g., attention distribution) in specific SI-VQA configurations.
> That said, the insights from [2] inspire exciting future research directions for our work. For instance, studying how the modality gap and cone effect evolve across the seven SI-VQA configurations could deepen our understanding of the interaction between embedding geometry and modality configurations. In addition, applying the techniques from [2] to modify the modality gap could serve as valuable ablation studies, helping assess how altering this gap influences VLM performance on SI-VQA tasks. We greatly appreciate the reviewer for highlighting this connection and pointing us toward opportunities for future research.
>
> [3] "From Recognition to Cognition: Visual Commonsense Reasoning"
>
> This paper introduces a large-scale dataset focused on visual commonsense reasoning, addressing biases such as answer-only biases through adversarial matching. Their objective is to create a robust dataset with questions, answers, and rationales, primarily targeting multi-choice Q&A settings affected by artifact annotations. In contrast, our study does not aim to construct a large-scale VQA dataset that is robust to such biases. Instead, we investigate modality biases by manipulating modality configurations in the SI-VQA dataset.
> Moreover, in our dataset, the model consistently attends to the question due to the task design, and we opted for close-ended questions over multiple-choice formats to minimize biases observed in early project experiments. Our objectives, therefore, differ significantly from [3], focusing on analyzing biases through controlled modality interplay rather than eliminating them in large-scale datasets.
>
> We are grateful to the reviewer for these suggestions, as they not only enrich our understanding of the field but also inspire directions for extending our work in future research. Thank you:)

---

### Official Review · Reviewer_gFMm · 2024-11-03

**Soundness:** 1
**Presentation:** 2
**Contribution:** 2
**Rating:** 3
**Confidence:** 4

**Summary:**

The paper investigates the limitations of Generative AI, particularly in Visual Language Models (VLMs), focusing on how the integration of image and text modalities affects performance in visual question answering (VQA) and reasoning tasks. They use only 100 samples to gain some conclusions in the paper, such as "complementary information between modalities improves answer and reasoning quality".

**Strengths:**

1. This paper is well-written and easy to understand.
2. The experimental analysis is comprehensive.
3. The conclusions drawn are intuitively credible.

**Weaknesses:**

1. The dataset contains only 100 samples, and the conclusions drawn lack novelty; they are basic findings that have been established in previous multimodal research. The importance of multimodal complementarity is widely recognized in the field, so the conclusions of this article lack originality.

2. Overall, this article is a fairly good technical report that provides a comprehensive experimental analysis.

**Questions:**

Are there any other interesting conclusions or exploratory directions for uncovering the importance of multimodal complementarity?

---

> ### Author Response · Authors · 2024-11-21
> **Rebuttal to Reviewer gFMm - PART 1/2**
>
> Thank you for your comments and for raising these important points. We appreciate the opportunity to clarify our findings and address your concerns.
>
> 1. The dataset contains only 100 samples, and the conclusions drawn lack novelty
>
> We thank the reviewer for their feedback. We acknowledge that the importance of multimodal complementarity is a well-established concept in the field. However, we believe our work brings a unique perspective and addresses gaps in the existing literature.
>
> **A Dataset for Model Behavior Analysis - Not A New Large-scale VLM Benchmark**
>
> Our dataset and methodology are not designed to benchmark large vision-language models (VLMs) but to probe their behavior in specific, controlled multimodal scenarios. This unique focus on modality roles and interactions represents an important step toward understanding the limits and strengths of VLMs.
>
> **Unique modality configurations**
>
> As discussed in the Related Work section, prior studies have indeed explored the role of modalities. However, these works often suffer from limitations such as data leakage (e.g., MMMU) or lack of complexity in the multimodal interaction (e.g., MMStar). For instance, MMStar involves a straightforward 4-option multi-class classification task without reasoning, minimal accompanying textual descriptions, and OCR-based dependencies. Additionally, the images in these datasets are not comparable or designed for controlled multimodal analysis. In addition, existing large-scale datasets often suffer from incomplete data, with missing ground-truth explanations (i.e., rationale) for vision-language model (VLM) answers. Moreover, images are not always strictly natural, with some containing pre-existing OCR data (text annotations) that introduce noise. Questions within these datasets vary widely in complexity, leading to inconsistency. In contrast, our curated dataset addresses these issues by featuring purely natural images across diverse domains, consistently complex questions, independent text descriptions, and no pre-existing annotations. This careful construction avoids confounding variables that could skew our results. Shaping unique modality configurations (or scenarios), we can analyze model performance, uncertainty, and attention mechanisms together, providing unprecedented insights into the interaction of modalities in a reasoning context.
>
> **Contribution to the Advancement of Vision Language Model Interpretability & Robustness**
>
> Understanding how modalities impact VLMs by exploring complex and specific scenarios is crucial as multimodal models become increasingly opaque and diverse. Our work demonstrates how task-specific modality configurations can reveal heterogeneous effects on model performance and uncertainty. This approach not only advances our understanding of VLMs but also highlights areas where these models require improvement to achieve robustness in diverse multimodal tasks.

---

> ### Author Response · Authors · 2024-11-21
> **Rebuttal to Reviewer gFMm - PART 2/2**
>
> **Regarding the dataset size,**
>
> -   **Evaluation purpose.** We emphasize that our dataset is only used for evaluation, not for training or fine-tuning models. Its purpose is to analyze pretrained models' behavior when faced with diverse input configurations.
>
> -   **TinyBenchmark.** While the SI-VQA dataset we designed remains small, prior recent research [[1]](https://arxiv.org/pdf/2402.14992) demonstrates that smaller datasets can still yield meaningful results when the quality of instances is high. Each instance of the SI-VQA Dataset underwent multiple rounds of refinement, ensuring that context descriptions serve only as background information without providing enough detail to answer the question independently.
>
> -  **Low Variance.** Despite the small size, the dataset exhibits low variance in model performance and uncertainty, as demonstrated in the Appendix H. Increasing the number of instances would not alter the conclusions drawn, as they hinge on the modality configurations rather than the dataset size. Instead, rebalancing the role of modalities, such as prioritizing text and introducing spurious visual information, would yield new insights---a direction we are actively exploring.
>
> -   **Scalability and Community Engagement.** To ensure data quality and consistency, we intentionally limited the dataset size to 100 well-curated instances. However, we have also developed a ready-to-use interactive interface that allows the community to generate additional instances with new images and textual descriptions. This ensures that our SI-VQA dataset is extensible and can grow with greater diversity and community input. See Appendix E for details about the Interface and instructions to create new data instances and the link to the code to try it out: https://gitlab.com/dekfsx1/isi-vlm.
>
> [1] [tinyBenchmarks: evaluating LLMs with fewer examples](https://arxiv.org/pdf/2402.14992)
>
> In summary, we respectfully argue that our work provides new insights into modality roles in VLMs and addresses a unique, underexplored aspect of multimodal research. We thank the reviewer for the opportunity to clarify our contributions and the broader context of our work.
>
> *We have modified the Introduction of the revised version of the paper to clarify the scope of the paper, novelties, and contributions. In the Discussion section, we elaborated more on the arguments about the reduced size. of the dataset*
>
> **Questions:**
> **Are there any other interesting conclusions or exploratory directions for uncovering the importance of multimodal complementarity?**
>
> **Other conclusion**: Our findings suggest that attention to a modality is not necessarily determined by the number of tokens associated with it. For instance, despite LLaVA-Next having approximately six times more image tokens, its attention toward those tokens remains comparable to models with fewer tokens -- like LLaVA-1.5.
>
> **Exploratory Direction**: A promising avenue for further research is investigating tasks where both modalities provide semantically distinct yet equally essential information. By analyzing scenarios where successful task completion (e.g., answering a question) requires complementary inputs from both modalities, we can deepen our understanding of how models integrate and prioritize multimodal information.

---

### Official Review · Reviewer_gZVH · 2024-11-05

**Soundness:** 3
**Presentation:** 3
**Contribution:** 3
**Rating:** 5
**Confidence:** 5

**Summary:**

This paper explores the impact of contextual information on Visual Question Answering (VQA) and reasoning within Vision-Language Models (VLMs). The study introduces the Semantic Interventions (SI)-VQA dataset and the Interactive Semantic Interventions (ISI) tool to evaluate how image and text modalities interact to affect model performance, accuracy, and uncertainty. The methodology involves benchmarking multiple VLM architectures under different configurations, integrating complementary or contradictory text with images. Experimental results indicate that integrating complementary information enhances model accuracy and reasoning quality, whereas contradictory information significantly degrades performance. Moreover, VLMs show a bias toward image inputs over textual context, with PaliGemma exhibiting notable overconfidence, leading to increased silent failures compared to LLaVA models. The study emphasizes the crucial role of modality integration and provides tools for better understanding VLM behavior in multimodal tasks.

**Strengths:**

1)	The impact of context and different modalities on VQA has always been a noteworthy topic. This paper's discussion, incorporating VLM, is insightful for future researchers.
2)	The experimental design in this paper is very thorough, with detailed consideration given to seven different input configurations.
3)	Some of the findings in the experimental results of this paper are very interesting and offer valuable insights for the design and application of future VLMs.
4)	This paper has released the dataset and the ISI tool.

**Weaknesses:**

1)	My primary concern is that the paper mainly describes the observed phenomena in the experimental results without providing sufficient analysis of why these results occur (though there is some experimental analysis). In particular, the paper does not explain how these results could be useful for advancing future VQA work or analyze what could be done to address some of the issues identified in the results. Additionally, some of the findings are not particularly novel, making the paper seem more like an experimental report.
2)	As the authors pointed out in the paper, the SI-VQA dataset has too few samples, with only one hundred entries. Although the authors believe these data are representative, they should at least analyze why the results from these one hundred samples are convincing. Is it because these one hundred samples are of high quality and diversity?

**Questions:**

1)	It appears that image text annotations have little effect on some of the model's metrics; for example, the results of Q+I_A+C_+ in Figure 2a are not optimal. Could the authors analyze the reason for this phenomenon?
2)	I don't quite understand why the initial hypothesis is introduced in Section 5.1, as it doesn't seem to be strongly related to the main part of the experiments.
3)	Could the authors explain specifically how GPT-4o is used as an evaluator of reasoning ability? Since the SI-VQA dataset has only 100 samples, why didn’t the authors consider using human evaluation instead? Would that provide more accurate results?

---

> ### Author Response · Authors · 2024-11-21
> **Rebuttal to Reviewer gZVH - PART 1/2**
>
> Thank you for your comments and for raising these important points. We appreciate the opportunity to clarify our findings and address your concerns.
>
> 1.  Providing sufficient analysis. [...]
>
> Despite the novelty aspects enumerated in the general answer to all reviewers, the conclusions of our paper are useful for advancing future VQA work:
>
> -   **Attention is not Explanation.** Attention distribution on input tokens does not correlate to VLM performance - see ablation study about using prompt engineering to steer model's attention toward the context and Figure 6. This conclusion from the paper contributes to the ongoing debate about the role of attention and whether it is a form of explanation. Our observations support the
>
> -   **Attention and Number of Tokens.** Our findings suggest that attention to a modality is not necessarily determined by the number of tokens associated with it. For instance, despite LLaVA-Next having approximately six times more image tokens, its attention toward those tokens remains comparable to models with fewer tokens -- like LLaVA-1.5.
>
> A promising avenue for further research is investigating tasks where both modalities provide semantically distinct yet equally essential information. By analyzing scenarios where successful task completion (e.g., answering a question) requires complementary inputs from both modalities, we can deepen our understanding of how models integrate and prioritize multimodal information.
>
> Could you please elaborate more to clarify what findings are not novel, and support that with prior work with similar findings? While our results might adhere to previous work, the uniqueness of the SI-VQA dataset and seven modality configurations offer a completely new context to evaluate modality interplay. While some findings were expected (role of complementary/contradictory context, ...), others are much more surprising and novel (role of annotations, the relation between attention and performance (see ablation study), effect of redundant information, the overconfidence of PaliGemma model,...).
>
> 2\.  Size of dataset.
>
> Regarding the dataset size,
>
> -   **Evaluation purpose.** We emphasize that our dataset is only used for evaluation, not for training or fine-tuning models. Its purpose is to analyze pretrained models' behavior when faced with diverse input configurations.
>
> -   **TinyBenchmark.** While the SI-VQA dataset we designed remains small, prior recent research [[1]](https://arxiv.org/pdf/2402.14992) demonstrates that smaller datasets can still yield meaningful results when the quality of instances is high. Each instance of the SI-VQA Dataset underwent multiple rounds of refinement, ensuring that context descriptions serve only as background information without providing enough detail to answer the question independently.
>
> -   **Low Variance.** Despite the small size, the dataset exhibits low variance in model performance and uncertainty, as demonstrated in the Appendix H. Increasing the number of instances would not alter the conclusions drawn, as they hinge on the modality configurations rather than the dataset size. Instead, rebalancing the role of modalities, such as prioritizing text and introducing spurious visual information, would yield new insights---a direction we are actively exploring.
>
> -   **Scalability and Community Engagement.** To ensure data quality and consistency, we intentionally limited the dataset size to 100 well-curated instances. However, we have also developed a ready-to-use interactive interface that allows the community to generate additional instances with new images and textual descriptions. This ensures that our SI-VQA dataset is extensible and can grow with greater diversity and community input. See Appendix E for details about the Interface and instructions to create new data instances and the link to the code to try it out: https://gitlab.com/dekfsx1/isi-vlm.
>
> [1] [tinyBenchmarks: evaluating LLMs with fewer examples](https://arxiv.org/pdf/2402.14992)
>
> *In the revised version of the paper, we integrated this argumentation in the Limitations and Future Directions paragraph in Discussion.*

---

> ### Author Response · Authors · 2024-11-21
> **Rebuttal to Reviewer gZVH - PART 2/2**
>
> Questions:
>
> 1.  It appears that image text annotations have little effect on some of the model's metrics; for example, the results of Q+I_A+C_+ in Figure 2a are not optimal. Could the authors analyze the reason for this phenomenon?
>
> We appreciate the reviewer's question regarding the limited effect of text annotations on the accuracy metrics.
>
> Our observations indicate that the performance consistently improves in the Q+I setting when including the annotations, but diminishes when context is added. Further, we observe that the models demonstrate higher attention to the annotated image compared to the original image (as shown in Fig. 5) and exhibit slightly lower semantic entropy in answering and reasoning, suggesting that the annotations are being processed but not consistently utilized to improve answering accuracy.
>
> To better understand this phenomenon, we examined the reasoning processes of the models to verify whether they observe and comprehend the annotations. Our findings reveal that while the LLaVA model rarely references text annotations during reasoning, the LLaVA-Vicuna and LLaVA-NeXT models frequently incorporate these annotations. For instance, the LLaVA-NeXT model reasons: "The image shows a beaker of blue liquid with a label that says 'Chemical solution.' It is not safe to drink this liquid without knowing its contents and potential hazards. [...]" This example demonstrates that these models are capable of recognizing and interpreting text annotations.
>
> However, even in cases where annotations are acknowledged during reasoning, the models often fail to utilize this information effectively in VQA answering, particularly in the presence of additional context. We observe that while the models can correctly recite annotations when directly queried, they struggle to generalize this information to answer the questions directly.
>
> We interpret this as a limitation in the models' ability to integrate multimodal information consistently, as the context still has large effect and the provided information is not utilized efficiently, only if the model is specifically asked to reason (this observation is, e.g., also made in chain of thought reasoning [1])
>
> [1] Wei, Jason, et al. "Chain-of-thought prompting elicits reasoning in large language models." Advances in neural information processing systems 35 (2022): 24824-24837.
>
> 2\. I don't quite understand why the initial hypothesis is introduced in Section 5.1, as it doesn't seem to be strongly related to the main part of the experiments.
>
> We introduce the first hypothesis to demonstrate the essential role of the image in our setting. The SI-VQA dataset has been designed so that the question cannot in theory be answered without the image modality. Validating the first hypothesis proves that the model does not rely on prior knowledge from pre-training (data leakage) to answer the question, avoiding limitations from previous datasets [1,2]
>
> [1] Yue, Xiang, et al. "Mmmu: A massive multi-discipline multimodal understanding and reasoning benchmark for expert agi." Proceedings of the IEEE/CVF Conference on Computer Vision and Pattern Recognition. 2024.
>
> [2] Chen, Lin, et al. "Are We on the Right Way for Evaluating Large Vision-Language Models?." arXiv preprint arXiv:2403.20330 (2024).
>
> 3\.  Could the authors explain specifically how GPT-4o is used as an evaluator of reasoning ability? Since the SI-VQA dataset has only 100 samples, why didn't the authors consider using human evaluation instead? Would that provide more accurate results?
>
> Appendix B details how GPT-4o is used as an evaluator of reasoning ability. In reasoning quality evaluation, GPT-4o is prompted to rate the reasoning from 0 to 10, using the prompt: "Rate the explanation's quality from 0 to 10. Give 10 for detailed, well-argued, and correct explanations. Give 0 for a poorly reasoned, wrong, or single-word explanation based on the question and image. Don't rate too harshly, use the full scale and output only the final score". During uncertainty computation, the number of the sampled outputs and the sampling temperature T are set to 10 and 0.9 respectively. We use conditional probabilistic sampling.
>
> As we designed the dataset, we also verified instance by instance the ratings given by GPT4-o for each reasoning. We have observed a very fair alignment with our own judgment and therefore decided to keep GPT4-o evaluation to measure the reasoning quality. We also plotted the distribution of reasoning quality scores by GPT-4o for all LLaVA architectures (see Figure 11 in Appendix H.1). We observed lower values for the standard LLaVA and a high bias towards the quality score of "8" for all models. The positive skewness is certainly due to the data diversity of ratings during GPT4-o training.

---

> ### Comment · Reviewer_gZVH · 2024-12-03
>
> Thank you to the authors for their patient responses, which have resolved some of my doubts. However, considering the overall quality of the paper, I will maintain my original rating.

---

### Official Review · Reviewer_MkZ9 · 2024-11-05

**Soundness:** 2
**Presentation:** 3
**Contribution:** 3
**Rating:** 6
**Confidence:** 5

**Summary:**

This work investigates the impact of different modalities – image and text – on the performance of Visual Language Models in Visual Question Answering (VQA) tasks.
The authors examine how the combination and interplay of these modalities affect accuracy, reasoning quality, model uncertainty, and attention attribution.
They collect a novel dataset (SI-VQA) with controlled interventions and an interactive tool (ISI) for manipulating image and text inputs to study VLM behavior.
This work sets the foundation for further analysis of modality integration in VQA, hightlighting the crucial role of context in guiding future model developments.

**Strengths:**

1. This work introduces the SI-VQA dataset, which is designed to require image-based answers, ensuring that visual content is essential for solving VQA tasks. This setup allows researchers to analyze how different modalities (image, text, context) influence the model’s accuracy, reasoning, and uncertainty​.

2. Comprehensive Benchmarking of VLMs: This work establishes a robust benchmark by evaluating various state-of-the-art VLMs under diverse modality configurations. This benchmarking approach highlights the contributions and limitations of each modality, as well as the strengths and weaknesses of different VLM architectures.

3. This work introduces the ISI Tool, enabling researchers to perform semantic interventions on VLM inputs, which supports fine-grained analysis of VLM behavior.

**Weaknesses:**

Please also refer to the questions section.

1. There are some concerns about the dataset.
2. The work lacks comparisons with other current datasets.
3. The work lacks supporting evidence for its claims.
4. The work lacks formal definitions of certain terms.

**Questions:**

Here are the corrected versions of the reviews:

1. The proposed dataset contains only 100 samples, which is quite limited in this domain.

2. The answers are limited to "Yes" or "No." Moreover, the paper does not specify the distribution of "Yes" versus "No" answers in the dataset. This leads to the following two concerns:

   - **Model Bias**: If the dataset is heavily skewed toward one answer (e.g., mostly "Yes" answers), it could introduce bias in the models, potentially leading them to favor that answer even when the visual information suggests otherwise.

   - **Impact of Interventions**: Without knowing the baseline distribution of answers, it is challenging to isolate the true effect of the semantic interventions (complementary context, contradictory context, image annotations) on the models' performance. For example, if the dataset already has a majority of "Yes" answers, an intervention that improves performance on "Yes" questions might not necessarily reflect a genuine improvement in the model's ability to understand the visual information.

3. Even though each sample is well-annotated (i.e., an image, a corresponding question with a ground truth Yes/No answer, a text-annotated version of the image, a contradictory context, and a complementary context), there are no comparisons between the proposed dataset and state-of-the-art (SOA) datasets regarding its advantages in Image-dependent Answers and Content Domain Diversity.

4. Regarding the claim that image text annotations have minimal impact on accuracy, or even decrease accuracy, the authors list some potential reasons for this, e.g., VLMs may already extract relevant information from images. It would be helpful to provide some qualitative or quantitative results to further support these explanations.

5. The term "modality relevance" is first mentioned in the abstract. However, there is no formal definition provided for it.

---

> ### Author Response · Authors · 2024-11-21
> **Rebuttal to Reviewer MkZ9 - PART 1/2**
>
> Thank you for your comments and for raising these important points. We appreciate the opportunity to clarify our findings and address your concerns.
>
> 1.  There are some concerns about the dataset. The proposed dataset contains only 100 samples, which is quite limited in this domain.
>
> Regarding the dataset size,
>
> -   **Evaluation purpose.** We emphasize that our dataset is only used for evaluation, not for training or fine-tuning models. Its purpose is to analyze pretrained models' behavior when faced with diverse input configurations.
>
> -   **TinyBenchmark.** While the SI-VQA dataset we designed remains small, prior recent research [[1]](https://arxiv.org/pdf/2402.14992) demonstrates that smaller datasets can still yield meaningful results when the quality of instances is high. Each instance of the SI-VQA Dataset underwent multiple rounds of refinement, ensuring that context descriptions serve only as background information without providing enough detail to answer the question independently.
>
> -   **Low Variance.** Despite the small size, the dataset exhibits low variance in model performance and uncertainty, as demonstrated in the Appendix H. Increasing the number of instances would not alter the conclusions drawn, as they hinge on the modality configurations rather than the dataset size. Instead, rebalancing the role of modalities, such as prioritizing text and introducing spurious visual information, would yield new insights---a direction we are actively exploring.
>
> -   **Scalability and Community Engagement.** To ensure data quality and consistency, we intentionally limited the dataset size to 100 well-curated instances. However, we have also developed a ready-to-use interactive interface that allows the community to generate additional instances with new images and textual descriptions. This ensures that our SI-VQA dataset is extensible and can grow with greater diversity and community input. See Appendix E for details about the Interface and instructions to create new data instances and the link to the code to try it out: https://gitlab.com/dekfsx1/isi-vlm.
>
> [1] [tinyBenchmarks: evaluating LLMs with fewer examples](https://arxiv.org/pdf/2402.14992)
>
> *In the revised version of the paper, we integrated this argumentation in the Limitations and Future Directions paragraph in Discussion.*
>
> 2.a. Answer distribution. The answers are limited to "Yes" or "No."
>
> Evaluating a large language model (LLM) with binary questions provides clear, unambiguous insights into its performance by simplifying the evaluation to a straightforward correct/incorrect framework. This reduces complexity compared to multiple-choice questions, where distractor quality and guessing can skew results, making binary questions ideal for precise assessments.
>
> *To further clarify our choice for the binary setting in this VQA and Reasoning task, we have added a justification in the revised version of the paper, Section 3.1.*
>
> 2.b. Moreover, the paper does not specify the distribution of "Yes" versus "No" answers in the dataset. This leads to the following two concerns: a. Model Bias, b. Impact of Interventions
>
> The paper specifies the distribution of "Yes" and "No" -- See **Appendix H.1. Figure 9,** Confusion matrices and accuracy values for all model architectures and configurations, shows a well-balanced answer distribution which addresses the reviewer's concern about model bias and impact intervention true effect measurement.

---

> ### Author Response · Authors · 2024-11-21
> **Rebuttal to Reviewer MkZ9 - PART 2/2**
>
> 3\. The work lacks comparisons with other current datasets. Even though each sample is well-annotated ..., there are no comparisons between the proposed dataset and state-of-the-art (SOA) datasets regarding its advantages in Image-dependent Answers and Content Domain Diversity.
>
> Given the highly specific setup of our dataset, which is unique in its design, making direct comparisons with state-of-the-art datasets is multifaceted. The SI-VQA dataset offers distinct advantages, particularly the unique combinations of image and text modalities.
>
> Regarding content domain diversity, we have sourced images from a wide range of domains, including geography, biology, household items, and sports, drawing from datasets such as MMMU and MMStar. This ensures that SI-VQA maintains a comparable diversity of topics to existing SOTA datasets.
>
> In terms of image-dependent answers, SI-VQA follows the example of MMStar by avoiding data leakage often found in datasets like ScienceQA, AI2D, MMMU, and SEED [3], and annotation artifacts where models can sometimes answer correctly without even considering the question [1]. However, unlike MMStar, which uses multiple-choice questions, we chose a binary question format (Yes/No) to eliminate issues related to distractor quality and random guessing, ensuring more precise evaluations.
>
> The complexity of questions in SI-VQA strikes a balance: it requires understanding object significance and simple interactions in the image without reaching the higher complexity levels of datasets like Visual Commonsense Reasoning [1] or Scene Graph VQA [2].
>
> Additionally, the ISI tool, partially used to design the SI-VQA dataset, provides exceptional flexibility to introduce new perturbations and explore the complementarity of text and image modalities. This capability enhances the complexity of comparing SI-VQA dataset to SOTA VQA and reasoning datasets in the future.
>
> In our revised version of the paper, we have extended Section 3.1 to account for a thorough comparison of SI-VQA to state-of-the-art datasets regarding its advantages in Image-dependent Answers and Content Domain Diversity and explanations for the design choices we made.
>
> [1] From Recognition to Cognition: Visual Commonsense Reasoning
>
> [2] GVQA: Learning to Answer Questions about Graphs with Visualizations via Knowledge Base
>
> [3] Are We on the Right Way for Evaluating Large Vision-Language Models?
>
> 4\. The work lacks supporting evidence for its claims. Regarding the claim that image text annotations have minimal impact on accuracy, or even decrease accuracy, the authors list some potential reasons for this, e.g., VLMs may already extract relevant information from images. It would be helpful to provide some qualitative or quantitative results to further support these explanations.
>
> The paper attributes the limited benefit of annotations for the models to information redundancy. We conduct a complementary analysis to demonstrate that redundant information can be either ineffective or even harmful to VLMs: the first ablation study introduces redundant textual information about the image. Our findings show that such redundant descriptions do not enhance model performance, reinforcing the claim that annotations do not improve predictions. As a matter of fact, when models are overloaded with redundant information, their performance may even decline, highlighting the uselessness or potentially detrimental impact of annotations or redundant textual descriptions.
>
> Given the ISI tool, we already planned to extend the SI-VQA dataset with an 8th-modality configuration where the image annotations are not guiding the model toward the correct answer, but rather misleading it. As this new configuration was first planned to observe how VLMs can manage contradicting information within the visual input, this is also an opportunity to bring additional quantitative results to show if "negative" annotations are accounted for, i.e., whether non-redundant OCR plays a significant role, as opposed to redundant "positive" annotations.
>
> 5\. The work lacks formal definitions of certain terms. The term "modality relevance" is first mentioned in the abstract. However, there is no formal definition provided for it.
>
> We acknowledge the confusion of that term and modified the abstract to change it to a well-defined concept that captures what we meant here: "attention attribution to modality"

---

### Author Response · Authors · 2024-11-21
**General Answer to Everyone - PART 1/3**

We sincerely thank all reviewers for their valuable comments. We appreciate the general consensus among the reviewers regarding the significance of our work for the community and the and the thoroughness of the experiments: "experimental design in this paper is very thorough" (Reviewer gZVH), "The experimental analysis is comprehensive and the conclusions drawn are intuitively credible" (Reviewer gFMm) or "there are some more dimensions that are considered by this paper [...]" (Reviewer 9N8L).

We acknowledge that the importance of multimodal complementarity is a well-established concept in the field. However, our work brings a unique perspective and addresses gaps in the existing literature.

Thus, next to the point-by-point responses to each reviewer, we will address the four main concerns here:

**tl;dr**\
The main criticism raised by all reviewers concerns the size of the SI-VQA dataset. Here, **(1)** we aim to clarify that SI-VQA was not intended to be a large-scale dataset for benchmarking VLMs. We will elaborate on why its limited size is appropriate and does not hinder the objectives and scope of this research. **(2)** We also want to emphasize our development of an interactive interface, the ISI tool (Appendix E), designed to engage the AI community in actively contributing to the expansion and diversification of the SI-VQA dataset. **(3)** Additionally, we emphasize the distinct focus of our work compared to previous studies addressing related topics, such as modality bias, VQA reasoning, and VLM interpretability. We highlight our contributions and novelties, demonstrating how our work advances research in explainable AI (XAI) and VLM interpretability. **(4)** Finally, we reiterate what our paper does not aim to achieve, ensuring that reviewers are clear on the objectives of our study.


1.  **Dataset Size**
Regarding the dataset size:

-   **Evaluation purpose.** We emphasize that our dataset is only used for evaluation, not for training or fine-tuning models. Its purpose is to analyze pretrained models' behavior when faced with diverse input configurations.

-   **TinyBenchmark.** While the SI-VQA dataset we designed remains small, prior recent research [[1]](https://arxiv.org/pdf/2402.14992) demonstrates that smaller datasets can still yield meaningful results when the quality of instances is high. Each instance of the SI-VQA Dataset underwent multiple rounds of refinement, ensuring that context descriptions serve only as background information without providing enough detail to answer the question independently.

-   **Low Variance.** Despite the small size, the dataset exhibits low variance in model performance and uncertainty, as demonstrated in the Appendix H. Increasing the number of instances would not alter the conclusions drawn, as they hinge on the modality configurations rather than the dataset size. Instead, rebalancing the role of modalities, such as prioritizing text and introducing spurious visual information, would yield new insights---a direction we are actively exploring.

-   **Scalability and Community Engagement.** To ensure data quality and consistency, we intentionally limited the dataset size to 100 well-curated instances. However, we have also developed a ready-to-use interactive interface that allows the community to generate additional instances with new images and textual descriptions. This ensures that our SI-VQA dataset is extensible and can grow with greater diversity and community input. See Appendix E for details about the Interface and instructions to create new data instances and the link to the code to try it out: https://gitlab.com/dekfsx1/isi-vlm.

[1] [tinyBenchmarks: evaluating LLMs with fewer examples](https://arxiv.org/pdf/2402.14992)

*In the revised version of the paper, we integrated this argumentation in the Limitations and Future Directions paragraph in the Discussion.*

---

### Author Response · Authors · 2024-11-21
**General Answer to Everyone - PART 2/3**

2. **ISI Tool**

The SI-VQA dataset comprises 100 carefully curated instances, but we want to emphasize the significant contribution of the interactive interface we developed: the ISI Tool (Appendix E). This tool allows users to design their own semantic interventions and create new benchmarks, complementing the SI-VQA dataset, which serves as the backbone for our sensitivity analysis with high-quality data.

As detailed in Appendix E, the ISI Tool enables users to load new images and questions, apply various semantic interventions (e.g., occlusions, annotations, shape additions), and craft custom text descriptions to guide or mislead the model to varying degrees. While this contribution may not have been highlighted by reviewers, we believe it is a major asset for the vision-language AI community. The tool facilitates further exploration of modality bias, modality importance, and the role of multimodal complementarity.

You can try the interface yourself at https://gitlab.com/dekfsx1/isi-vlm to see how it represents a novel and valuable addition to the field. Notably, our previous tool-focused paper was accepted at six NeurIPS workshops, demonstrating the community's enthusiasm for such tools that empower collaborative contributions to enriching datasets and advancing understanding in the field.

3. **Novelty and Contribution**

**A Dataset for Model Behavior Analysis - Not A New Large-scale VLM Benchmark**

Our dataset and methodology are not designed to benchmark large vision-language models (VLMs) but to probe their behavior in specific, controlled multimodal scenarios. This unique focus on modality roles and interactions represents an important step toward understanding the limits and strengths of VLMs.

**Unique modality configurations**

Prior studies have explored modalities but face limitations like data leakage (e.g., MMMU) or oversimplified multimodal tasks (e.g., MMStar). MMStar, for instance, involves basic multi-class classification with minimal reasoning, sparse textual descriptions, and OCR-based dependencies. Many large-scale datasets also lack complete data, including ground-truth rationales for VLM answers, and feature noisy or annotated images that hinder controlled analysis. Question complexity often varies, creating inconsistency. In contrast, our curated dataset addresses these issues with natural, annotation-free images, consistently complex questions, independent text descriptions, and well-defined modality scenarios. This enables robust analysis of model performance, uncertainty, and attention mechanisms in reasoning contexts.

**Contribution to the Advancement of Vision Language Model Interpretability & Robustness**

Understanding how modalities impact VLMs by exploring complex and specific scenarios is crucial as multimodal models become increasingly opaque and diverse. Our work demonstrates how task-specific modality configurations can reveal heterogeneous effects on model performance and uncertainty. This approach not only advances our understanding of VLMs but also highlights areas where these models require improvement to achieve robustness in diverse multimodal tasks.

**Other contributions to the Advancement of the explainable AI community**

Despite the novelty aspects enumerated in the general answer to all reviewers, the conclusions of our paper are useful for advancing future VQA work:

-   **Attention is not Explanation.** Attention distribution on input tokens does not correlate to VLM performance - see ablation study about using prompt engineering to steer model's attention toward the context and Figure 6. This conclusion from the paper contributes to the ongoing debate about the role of attention and whether it is a form of explanation. Our observations support the

-   **Attention and Number of Tokens.** Our findings suggest that attention to a modality is not necessarily determined by the number of tokens associated with it. For instance, despite LLaVA-Next having approximately six times more image tokens, its attention toward those tokens remains comparable to models with fewer tokens -- like LLaVA-1.5.

**Future directions to address observations made in the paper.**

A promising avenue for further research is investigating tasks where both modalities provide semantically distinct yet equally essential information. By analyzing scenarios where successful task completion (e.g., answering a question) requires complementary inputs from both modalities, we can deepen our understanding of how models integrate and prioritize multimodal information.

---

### Author Response · Authors · 2024-11-21
**General Answer to Everyone - PART 3/3**

4. **What our paper is not**

To prevent any confusion about the nature and scope of our research paper, we would like to remind reviewers and Area Chair what our paper is not about:

1\. We are not proposing a new large-scale multimodal benchmark aimed at achieving state-of-the-art results in multimodal learning.

2\. We are not attempting to resolve modality bias.

In summary, our work provides new insights into modality roles in VLMs and addresses a unique, underexplored aspect of multimodal research. We thank the reviewer for the opportunity to clarify our contributions and the broader context of our work.

*We have modified the Introduction of the revised version of the paper to clarify the scope of the paper, novelties, and contributions. In the Discussion section, we elaborated more on the arguments about the dataset's reduced size.*

For more detailed changes, see the reviewer's individual answers.

---

### Author Response · Authors · 2024-11-25
**Rebuttal Deadline Reminder**

As the discussion phase is approaching its end soon, we wanted to kindly check if you have any remaining points for us to address. If we do not receive any further comments on our rebuttal, we assume that all points of criticism have been addressed appropriately.
We appreciate your valuable feedback and are happy to clarify anything further if needed.

---

### Comment · Area_Chair_pu2v · 2024-12-02
**Dec 2: the last day for reviewers to ask questions to authors**

Dear Reviewers,

Thank 9N8L for confirming the authors' responses and update the review.

Dear MkZ9, gFMm, gZVH, as Dec 2 is the last day for reviewers to ask questions, can you take this chance to check authors' responses and other reviewers' comments and confirm whether you have any remaining concerns? Your constructive discussion and suggestions would be great contributions to the review process.

Best,

AC

---

### Meta-Review · Area_Chair_pu2v · 2024-12-20

**Metareview:**

Summary: This work discussed about the impact of image and text on VLMs for VQA tasks, especially the impacts on accuracy, reasoning quality, model uncertainty, and attention attribution. A high-quality dataset SI-VQA and an interactive tool (ISI) are provided to study VLM behavior.

Main Strengths: (1) The impact of context and different modalities on VQA is important research question. (2) The experimental design is very thorough. (3) The Interactive Semantic Interventions (ISI) tool is introduced to support further analysis of VLMs.

Major Weaknesses & Concerns: (1) The proposed Semantic Interventions (SI-)VQA dataset only contains 100 instances. (2) The answers are limited to binary answers, yes and no. (3) Experimental analysis is not strong or insightful enough for future network designs, and validness of the experimental analysis is not clear, which limits the contributions of the work.

This paper received ratings as 6, 5, 5, 3. AC read reviewers' comments and authors' responses, exclude short review for final decisions, and think the current version of the paper is borderline. Considering the commonly shared concern on scale of dataset, the AC does not recommend accepting the paper and encourage the author to further improve the quality and quantity of evaluation dataset for more convincing analysis.

**Additional Comments On Reviewer Discussion:**

During the discussion, the major and common concerns is that the scale of dataset is too small (only 100 instances).

---

### Decision · Program_Chairs · 2025-01-22

Reject